# New Late Pleistocene age for the *Homo sapiens* skeleton from Liujiang southern China

Junyi Ge [1,2,14], Song Xing [1,3,14], Rainer Grün[4,5,6], Chenglong Deng[2,7], Yuanjin Jiang[8], Tingyun Jiang[6], Shixia Yang [1], Keliang Zhao [1], Xing Gao[1,2], Huili Yang[9], Zhengtang Guo [10], Michael D. Petraglia [5,11,12] ✉ & Qingfeng Shao [6,13] ✉

The emergence of *Homo sapiens* in Eastern Asia is a topic of significant research interest. However, well-preserved human fossils in secure, dateable contexts in this region are extremely rare, and often the subject of intense debate owing to stratigraphic and geochronological problems. Tongtianyan cave, in Liujiang District of Liuzhou City, southern China is one of the most important fossils finds of *H. sapiens*, though its age has been debated, with chronometric dates ranging from the late Middle Pleistocene to the early Late Pleistocene. Here we provide new age estimates and revised provenience information for the Liujiang human fossils, which represent one of the most complete fossil skeletons of *H. sapiens* in China. U-series dating on the human fossils and radiocarbon and optically stimulated luminescence dating on the fossil-bearing sediments provided ages ranging from ~33,000 to 23,000 years ago (ka). The revised age estimates correspond with the dates of other human fossils in northern China, at Tianyuan Cave (~40.8–38.1 ka) and Zhoukoudian Upper Cave (39.0–36.3 ka), indicating the geographically widespread presence of *H. sapiens* across Eastern Asia in the Late Pleistocene, which is significant for better understanding human dispersals and adaptations in the region.

Dating of hominin fossils and their localities places the origin of *Homo sapiens* back to ~310 ka in Africa[1] and documents dispersals out of Africa and into the Levant and southeastern Europe to more than ~180 ka[2,3]. Recent advances in understanding the biological and cultural evolution of modern human populations[2–4] highlight the need to clarify the history of *H. sapiens* in the East Asian mainland. However, fossils of *H. sapiens* in this region with clear stratigraphic contexts and precise chronological age controls are scarce. Though

[1]Key Laboratory of Vertebrate Evolution and Human Origins, Institute of Vertebrate Paleontology and Paleoanthropology, Chinese Academy of Sciences, Beijing 100044, China. [2]University of Chinese Academy of Sciences, Beijing 100049, China. [3]Centro Nacional de Investigación sobre la Evolución Humana, Paseo de la Sierra de Atapuerca s/n, Burgos, Spain. [4]Research School of Earth Sciences, The Australian National University, Canberra, ACT, Australia. [5]Australian Research Centre for Human Evolution, Griffith University, Brisbane, QD 4111, Australia. [6]School of Geography, Nanjing Normal University, Nanjing 210023, China. [7]State Key Laboratory of Lithospheric Evolution, Institute of Geology and Geophysics, Chinese Academy of Sciences, Beijing 100029, China. [8]Lotus Cave Science Museum, Liuzhou 545001, China. [9]State Key Laboratory of Earthquake Dynamics, Institute of Geology, China Earthquake Administration, Beijing 100029, China. [10]Key Laboratory of Cenozoic Geology and Environment, Institute of Geology and Geophysics, Chinese Academy of Sciences, Beijing 100029, China. [11]Human Origins Program, National Museum of Natural History, Smithsonian Institution, Washington, DC 20560, USA. [12]School of Social Science, The University of Queensland, Brisbane, QD 4072, Australia. [13]Key Laboratory of Virtual Geographic Environment, Ministry of Education, Nanjing Normal University, Nanjing 210023, China. [14]These authors contributed equally: Junyi Ge, Song Xing. ✉e-mail: m.petraglia@griffith.edu.au; qingfengshao@njnu.edu.cn

there have been claims for *H. sapiens* in China before 80 ka[5–7], the taxonomy and ages of these finds have been contentious[8,9]. More secure knowledge of early modern human history in the East Asian mainland rests on remains from Tianyuan Cave (40.8–38.1 ka cal BP)[10] and the casts of the lost fossils from Zhoukoudian Upper Cave (39.0–36.3 ka cal BP)[11] in northern China, and by young fossils of Pleistocene-Holocene transition from Longlin cave (11.5 ka cal BP)[12], Maludong (14.3–13.6 ka cal BP)[12], and Dushan cave (15.9–12.8 ka cal BP)[13] in southern China.

Among key fossil evidence in China, the Liujiang hominin has assumed great importance owing to the fact it is one of the most complete *H. sapiens* skeletons. Moreover, the skeletal remains are thought to represent a late Middle Pleistocene/early Late Pleistocene presence of modern humans in Eastern Asia, even earlier than their arrival in western Eurasia. Though the Liujiang skeletal elements are generally well preserved, the provenience and dating of the fossils have been contentious since they were first found more than six decades ago[14–16].

The Liujiang materials were originally recovered in September 1958, in a cave called Tongtianyan by workmen digging for phosphorous fertilizer[17] (Fig. 1). After the human fossils were discovered, paleoanthropologists from the Institute of Vertebrate Paleontology and Paleoanthropology (IVPP), Beijing, immediately travelled to the cave to conduct initial investigations. The human fossils were found near the entrance of the cave (Fig. 1c). The surveyors found that only a small part of the deposits were still intact, consisting of limestone breccia and unconsolidated sediments, including a layer in which vertebrate fossils were recovered. Unfortunately, the exact position of the human fossils was unclear given the intensity of the digging by the work crews. At the time it was noted that the human cranium was likely embedded in unconsolidated breccia, which was markedly different from the consolidated deposits containing the vertebrate fossils. In addition, no artefacts have ever been recovered from Tongtianyan, indicating that the cave was not an occupation site.

The Liujiang human fossils are composed of a nearly complete cranium and 17 postcranial elements, including vertebrae, ribs, a sacrum, the os coxae, and femora[17]. The cranial volume is 1567 mL. The right and left zygomatic arches are broken. The teeth and palate are moderate in size, with a shovel-shaped right lateral incisor[17]. There is a

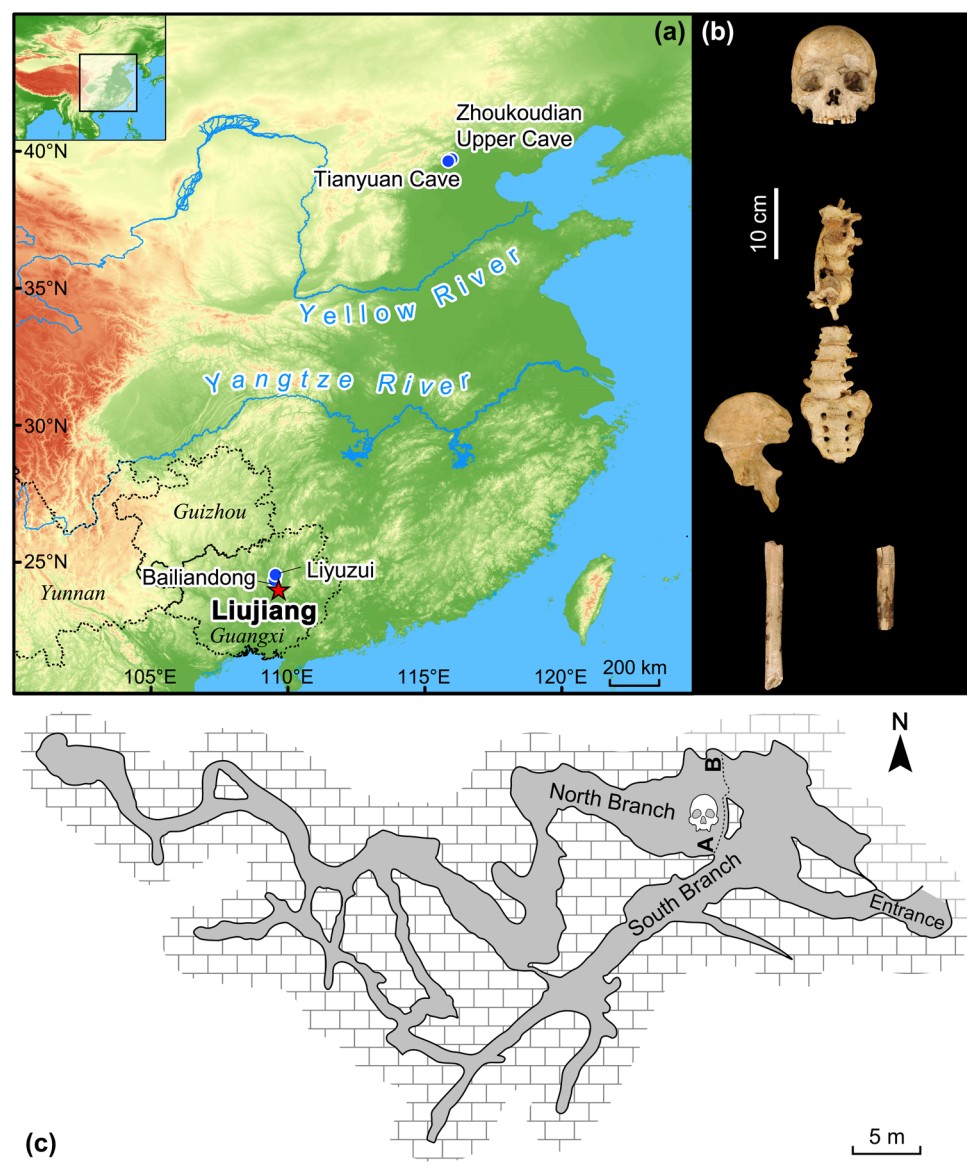

**Fig. 1 | Map and plan view of Tongtianyan cave. a** Location of Tongtianyan cave (Liujiang) in Guangxi Province, southern China, together with the location of other key fossils of *Homo sapiens* in China. **b** Frontal view of the Liujiang cranial and postcranial elements. **c** Plan map of Tongtianyan cave. A–B in this panel refers to the depositional section shown in Fig. 2a.

congenital absence of both upper third molars and the loss of the upper left central incisor and upper left second molar crowns. The vertebrae include the inferior four (the ninth to twelfth) thoracic vertebrae and all five lumbar vertebrae. Four articulating rib fragments are attached to the thoracic vertebrae. The sacrum is still attached to the fifth lumbar vertebra and could articulate with the right os coxae by the auricular surface. The right femur is represented by the diaphysis of the most proximal to middle-distal region while the left one is represented by the diaphysis of the proximal to middle region.

The Liujiang materials were attributed to a single individual based on the similar colour of the bones, the similar degree of fossilization, and the lack of duplication of anatomical parts[17]. The degree of cranial suture closure and the dental occlusal wear indicate that the individual was an adult, approximately 40 years old[17]. The upper two-thirds of the sacrum is relatively straight with the tail being curved anteriorly. The sciatic notch has a relatively small posterior potion. The characteristics of the os coxae, in combination with the cranial features suggest that the individual is a male[17,18].

Human fossils are generally rare and/or partial across East and Southeast Asia. For instance, although Tianyuan Cave preserves postcranial bones, the cranium is absent[13]. Salkhit is represented only by a skullcap, Tam Pa Ling has two mandibles, an incomplete cranium and other postcranial fragments, and the Deep Skull from Malaysia's Niah Cave is also incomplete[19–23]. The human remains from the Tabon Cave in Philippines have cranial fragments, two partial mandibles, and other limb fragments[24,25]. The Wajak materials from Indonesia are mainly known for a relatively complete cranium and a partial skull[26]. In contrast to these fragmentary fossil finds, the Liujiang individual retains a complete cranium, as well as vertebrae, ribs, femora, and pelvic bones that are absent at several other sites. As one of the most complete fossils of *H. sapiens* in China, the Liujiang skeleton provides crucial information about the morphology and evolution of modern humans in Eastern Asia. The Liujiang individual exhibits a series of modern human-like features, such as a globular cranium, a reduced and flat face, a round braincase with an enlarged cranial volume, small teeth with simple occlusal surfaces, and a slim body shape[17,27–29]. Among the cranial features, the short and broad face and the rectangular orbit are the traits that are rarely expressed in recent modern humans, but commonly found in early modern human fossils[17,30,31]. In comparison to other Eurasian early modern humans, Liujiang has some distinct features, including a bulged frontal squama, protruded superciliary arches, a low nasal bridge, and an occipital bun[32]. There is a general agreement that the Liujiang cranium can be separated from that of modern human finds from Zhoukoudian Upper Cave in both metrics and non-metrics[30,31,33]. In comparison to Liujiang, the cranium of Zhoukoudian Upper Cave retains more primitive and robust characteristics. As a derived form, the Liujiang cranium is more rounded and has a less developed supraorbital torus, sagittal keel, occipital torus, mastoid process, zygomatic triangle, and zygomatic tubercle than that of Zhoukoudian Upper Cave[31,32]. A regional population affinity for Liujiang was based on features including facial flatness, the orientation of the broad and low nose, rounding of the lower outside borders of the orbits, large and protrusive zygoma, shoveling of the incisors, and a congenital absence of a third molar[17,28,34]. However, this point of view was challenged by a craniometric analysis that demonstrated that the Liujiang individual could be distinguished from modern regional populations[30]. On the basis of other craniometric studies, the Liujiang cranium was shown to be morphologically divergent from Moh Khiew, Thailand, dated to 25.8 ka cal BP[35], but similar to Minatogawa 1, Japan, dated to ~18.3–16.6 ka cal BP[36]. A geometric morphometric study of cranial shape indicated that Liujiang was similar to Cro-Magnon 1[32], dating to ~33–31 ka cal BP[37].

Since the discovery of the hominin fossils, two independent radiometric dating projects have been conducted[14–16]. Using conventional radiocarbon and classic α-counting U-series dating methods,

Yuan and colleagues[15] obtained a [14]C age of 3.0 ± 0.2 ka cal BP for the flowstone near the cave entrance, a U-series age of 67 + 6/−5 ka for the thick flowstone on top of Unit II (see Supplementary Information, section 1.2), and U-series ages ranging between 227 and 95 ka for the mammalian fossils. The investigators suggested that the human fossils were older than ~67 ka, but they noted that this age estimate remained to be verified given the uncertain provenience of the finds. A later attempt to establish the age of the Liujiang deposits was by Shen and colleagues[14,16] using α-counting and thermal ionization mass spectrometry U-series methods to date the flowstones from various depositional units and mammalian fossils. The researchers concluded that the human fossils dated to at least ~68 ka and more likely to ~139–111 ka if they came from the refilling breccia. Given the combination of early ages, the Liujiang skeletal remains have been considered to be among the earliest fossil representatives of *H. sapiens* in China, and therefore, the hypothesis of regional continuity of human populations in the area was thought to be strengthened[38]. Yet, the late Middle Pleistocene to early Late Pleistocene dating results have been considered surprisingly old given the modern morphological features of the human fossils[31].

Tongtianyan cave (24°10′59″N, 109°25′56″E, 164 m above mean sea level) is situated on the western slope of a karstic limestone mountain in Liujiang District of Liuzhou City, ~180 km northeast of Nanning, the capital of the Guangxi Province. Geological and stratigraphic observations indicate that the sedimentary sequence in the cave can be divided into three major units (I–III) that are irregularly bedded from bottom to top (Fig. 2). Unit I is composed of a light gray tilt-bedded fine sand interbedded with thin layers of yellow clay covering the cave bottom and irregularly underlying Units II and III. Unit II, approximately 3.6 m-thick, consists primarily of yellowish-brown bedded calcareous clay/silty clay and silty clay interbedded with several flowstone layers. Most of Unit II is exposed along the north side of the cave, with a small amount on the south side. Well-preserved mammal species fossils were recovered from Unit II[14,17] including *Ailuropoda melanoleucus, Rhinoceros sinensism, Stegodon orientalis, Megatapirus* sp*., Sus* sp., Bovidae and Cervidae. These fossils are the common members of the late Middle Pleistocene *Ailuropoda-Stegodon* fauna of southern China[39,40]. Unit III has a thickness of 3.8 m and mainly consists of brown gravel and yellowish-brown clay capped by a flowstone layer. It fills in the gully cutting through Unit II, part of which irregularly overlies the upper part of Unit II. Unit III can be subdivided into five layers from top to bottom (Supplementary Information, section 2). The yellowish-brown clay sediments cemented on the human fossils suggest that Unit II and upper part of Unit III could be the potential human-fossil bearing stratigraphy, rather than the coarse breccia layer as previously suggested[14].

The previous early age estimates of Liujiang skeletal remains have assumed great importance in the paleoanthropological record of China, with significant implications for human evolution and the timing of out of Africa dispersals[41,42]. Precise dating for the Liujiang skeletal remains is therefore crucial for determining its evolutionary position and disentangling the long-standing debate on the timing of modern human occupation of the East Asian mainland. Here, we show detailed sedimentological and geochemical comparisons between the sediments from the Liujiang human fossils and those from various depositional units at Tongtianyan cave to determine the origin of the human specimens. Subsequently, we combine [14]C, U-series and optically stimulated luminescence (OSL) dating of the human and mammalian fossils, and sedimentary deposits to provide age constraints on this critical early modern human skeleton.

## Results
### Sediment characteristics and provenience of the human fossil remains
To pinpoint the provenience of the Liujiang human fossils, sediment samples were extracted from the medullary cavity of the Liujiang

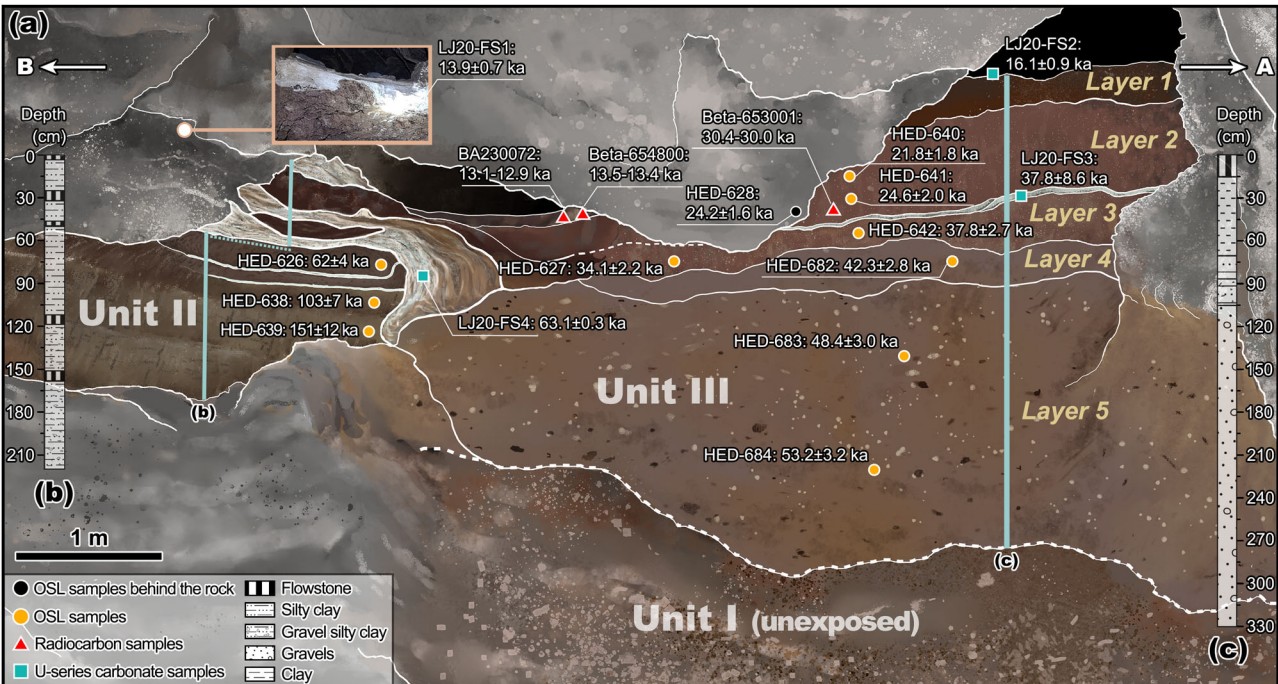

**Fig. 2 | Tongtianyan cave stratigraphy with chronological age estimates. a** Unit II and Layers 1–5 of Unit III: OSL (filled circles), radiocarbon (filled triangles), carbonate U-series dates (filled squares). **b, c** Lithostratigraphy of the columns refers to the blue vertical bars in the sedimentary sequences of Units II and III. A and B with arrows in the upper left and right sides correspond to the transect illustrated in Fig. 1c.

femur for Sr-Nd isotopic (Supplementary Table 1), trace element (Supplementary Data 1), grain-size analyses and colour assessment, allowing for a comparison to other depositional units within the cave. The Sr-Nd isotopic composition of the sediments from the human fossils (LJHS) clearly differed from the sediments of Unit II and Layer 5 of Unit III (Fig. 3a, Supplementary Table 1), thereby excluding these as a potential provenience for the fossils. The rare element ratio plots and ternary diagrams (Supplementary Fig. 5b–i) show that the LJHS results are most similar with the samples from Layers 2 and 3 of Unit III, but significantly different to those from Unit II and Layer 5 in Unit III. Sediment grain-size and colour analyses also show that the LJHS data are most closely related to Layer 2 of Unit III (Fig. 3b, c, Supplementary Fig. 3). Thus, multiple lines of evidence point to the reddish-brown clay of Layer 2 of Unit III as the burial environment for the human fossils. The Layer 5 of Unit III, previously proposed as the origin context[14,16], can categorically be excluded unless the fossils have been reworked and redeposited. The articulation of the vertebrae, with no trace of gnawing or abrasion on the human fossils, seems to exclude the possibility that the human fossils were buried in the sandy gravel layer or underwent post-depositional transport; instead, the articulations favour an interpretation of burial in a low-energy depositional environment without significant transport.

## Dating results

We then established a chronologic framework for the stratigraphy based on radiocarbon, OSL and U-series dating (see Supplementary Materials 4, 5 and 6). U-series dating on the capping flowstone, Layer 1 of Unit III, yielded ages of ~16–14 ka (Supplementary Fig. 6, Supplementary Data 2), similar with the radiocarbon dates of ~14–13 ka cal BP obtained from charcoal and organic sediment in this layer (Supplementary Table 2). U-series dating on the flowstone on the top of Unit II resulted in an age of ~63 ka (Supplementary Fig. 6, Supplementary Data 2), consistent with previously published results[14]. Quartz OSL dating results of the samples from Unit III ranged from ~53 to 22 ka (2σ uncertainty range), which are stratigraphically coherent and broadly

consistent with the U-series (Fig. 2, Supplementary Data 2) and radiocarbon ages (Fig. 2, Supplementary Table 2). Due to the saturation of the luminescence signals, samples from the upper and lower parts of Unit II provided only minimum ages of at least 54 ka and 106 ka, respectively.

The U-series results from the flowstones, the radiocarbon and the OSL age estimates from Unit III (Supplementary Data 3 and Supplementary Data 4) were combined with stratigraphic information to develop a Bayesian age model (Fig. 4a, b, Supplementary Code 1). Details of samples, preparation, measurement, and data-analysis procedures are in Supplementary Materials (section 6.2). The Bayesian analyses indicate that Unit III started to accumulate from 52.2 ± 11.0 ka (henceforth all age estimates are given with 2σ uncertainties) with an end date of 11.6 ± 1.6 ka. This is supported by the U-series results of the thickest flowstone overlying Unit II (63.1 ± 0.3 ka) and on the capping flowstones of Unit III (Layer 1, ~16–14 ka). Layers 3 and 2 span from 39.4 to 37.2 ka and from 32.5 to 22.6 ka, respectively. The boundary between Layers 3 and 2 was modelled from 37.2 ± 5.2 ka to 32.5 ± 2.5 ka (Fig. 4b, Supplementary Data 5), consistent with the U-series isochron age of 37.8 ± 8.6 ka obtained from a dirty flowstone sample (LJ20-FS3) located between Layers 2 and 3 (Supplementary Fig. 6, 7). Therefore, the best age estimate for the fossil-bearing Layer 2 is between 32.5 ka and 22.6 ka.

Fossil bone readily takes up uranium from ground water post deposition although the process can be relatively slow; therefore, U-series dating of fossil bone allows estimation of a minimum burial age, provided U-leaching has not occurred[43]. U-series dating analyses on the Liujiang human fossils were first carried out on two small dentine fragments fallen from the upper central incisor, and some bone splinters produced during a previous sample extraction from the left femur (Figs. 5a-1–3). The dentine sample yielded U-series apparent age of 18.72 ± 0.10 ka, and the two bone sub-samples 22.85 ± 0.04 ka and 21.64 ± 0.04 ka, respectively. The difference between the dentine and bone U-series ages (~3–4 ka) is likely the result of the fact that the dentine was in an open system for a longer period of time.

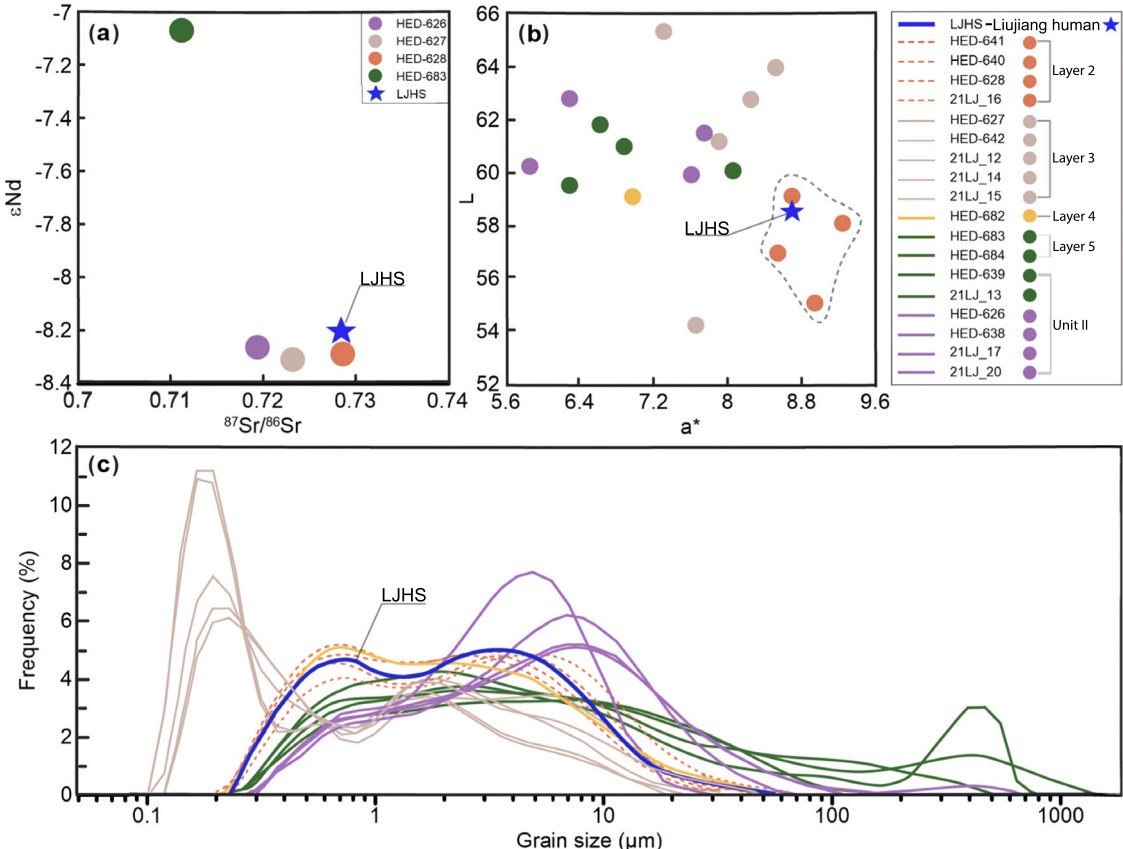

**Fig. 3 | Tongtianyan cave grain-size distributions and provenience information.**
**a** Nd-Sr diagram. **b** colour parameters. **c** Grain-size distributions for sediments from the Liujiang stratigraphic sequence and the sediments housing in the medullary cavity of the Liujiang left femur (LJHS). a* (L) refers to the redness (lightness) value of these sediments. Source data for this figure are provided as a Source Data file.

We further undertook U-series analyses along two profiles on the left femur to check for diffusion processes. Two parallel profiles consisting of nine and eight subsamples (LJ20-1 to 9 and LJ20-(1) to (8)), respectively, were hand-drilled sequentially from the inner to the outer surfaces on the cortical bone section (-6 to 7 mm thick, Figs. 5a-4, 5). U-series analyses show that the $^{234}U/^{238}U$ and $^{230}Th/^{238}U$ activity ratios are in the narrow ranges of 1.189–1.195 and 0.218–0.228, respectively, and U-series apparent ages are between 21.4 and 23.3 ka (Figs. 5a-5). The U-series isotopic ratios and apparent ages all display U-shaped profiles across the bone section (Fig. 5b), which conform to the distribution patterns predicted by the D-A (diffusion and adsorption)[44,45] and DAD (diffusion-adsorption-decay) models[43]. For the two profiles, the DAD model yielded consistent results: maximum likelihood minimum ages of 23.8 ± 0.7 and 24.0 ± 0.8 ka with initial $^{234}U/^{238}U$ activity ratios of 1.192 and 1.191. Therefore, we consider that the Liujiang human fossils have a burial age of at least 23.9 ± 0.5 ka.

Tongtianyan cave has long been seen as a site demonstrating the co-existence of *H. sapiens* alongside *Ailuropoda-Stegodon* fauna during the Late Pleistocene of south China[34], reinforcing evidence for the deep antiquity of the human fossils. Previous U-series dating of seven fossil teeth ranged from 227 to 95 ka[14,16]. We collected eight mammalian fossil teeth for U-series dating to test this relationship. Among them, one fossil tooth was recovered in the sediments adhering to the cave ceiling and seven from the disturbed sediments as a consequence of digging for fertilizer (Supplementary Fig. 8, Supplementary Data 2). The newly sampled mammalian fossil teeth gave $^{234}U/^{238}U$ and $^{230}Th/^{234}U$ activity ratios and apparent ages comparable to the previous results (Figs. 5c-1, Supplementary Data 2). However, the human fossils were distinct from the mammalian fauna in terms of U-series isotopic ratios and apparent U-series ages (Fig. 5c), indicating that there is no association between the human fossils and the *Ailuropoda-Stegodon* fauna.

In sum, provenience and dating studies have been conducted on the flowstone, sediments and the human and mammalian fossil remains. Our proveniencing results indicate that the Liujiang human fossils derived from Layer 2 of Unit III. Layer 2 ranged from 32.5 ± 2.5 to 22.6 ± 7.4 ka using Bayesian analysis on radiocarbon, OSL and carbonate U-series ages. U-series dating on the human fossils provided a minimum age of 23.9 ± 0.5 ka, falling into the age range for Layer 2. Collectively, the age of the Liujiang human fossils can be constrained to ~33–23 ka. The mammalian fossils dated to between 227 ka and 95 ka by U-series methods[14-16] indicate a significant hiatus between the deposition of the *Ailuropoda-Stegodon* fauna and the human remains.

## Discussion

Since its discovery in 1958, the Liujiang skeletal remains have been considered as one of the most significant human fossils from Eastern Asia, and owing to the excellent preservation, the cranial, dental and postcranial remains have been the subject of a number of biological and morphological comparisons across Eurasia[17,18,30,31,46]. The age of the Liujiang skeletal remains has great importance for understanding human dispersals to Eastern Asia and the occupation history of China[46]. Previous dating of the Tongtianyan cave deposits to >150 ka or 139–111 ka by Shen et al.[14] and to >67 ka by Yuan et al.[15] suggested that the Liujiang skeletal remains represented an early presence of *H. sapiens* in China, ranging sometime between the late Middle Pleistocene to MIS (Marine Isotope Stage) 5. Following the publication of these old ages[14], the Liujiang skeletal remains emerged as one of the earliest modern humans outside of Africa, even older than Qafzeh and Skhul from Israel, and predating the modern human occupation of

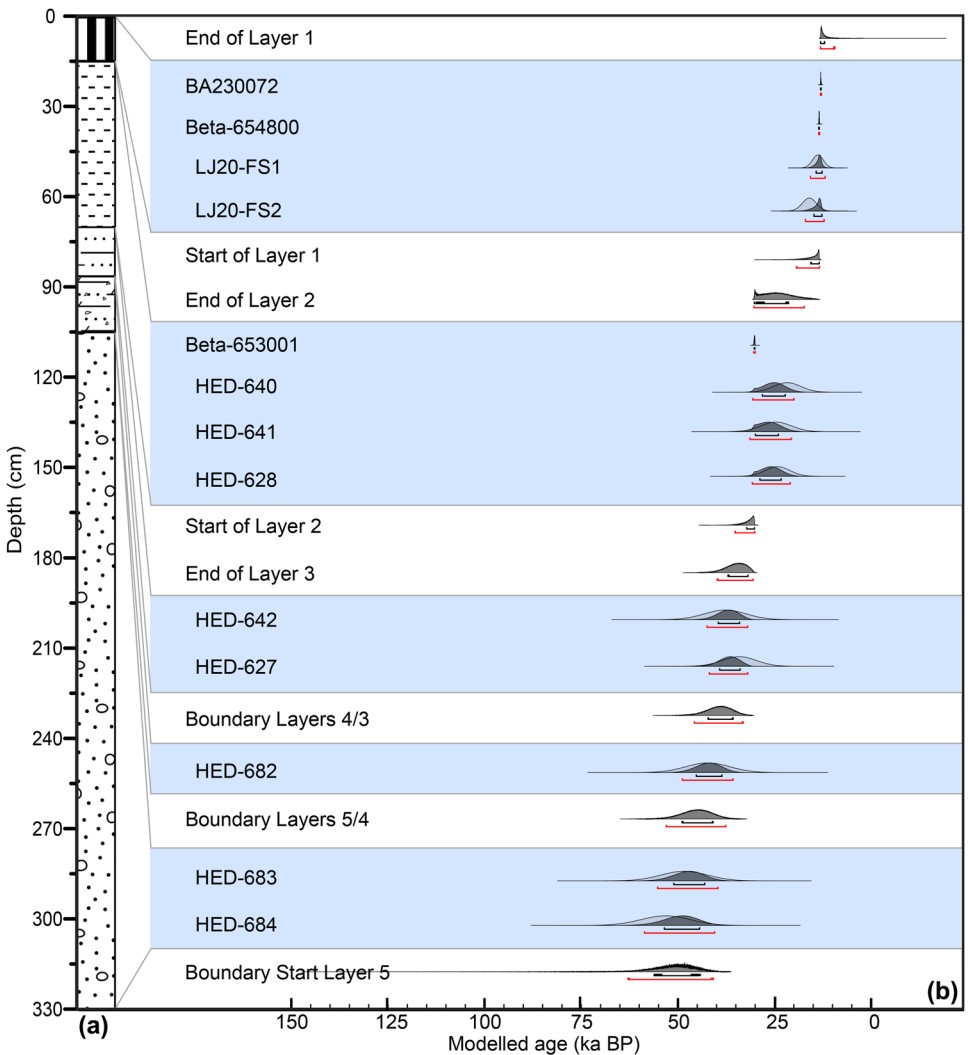

**Fig. 4 | Tongtianyan cave Unit III stratigraphy with Bayesian age model.** The black and red bars beneath each distribution represent the 68.3 and 95.4% probability ranges of the modeled ages (**b**). The lithostratigraphy of the columns in (**a**) refers to that in Fig. 2c.

Europe. Consequently, the early appearance of *H. sapiens* in Eastern Asia implied a different pattern of origin. Our dating study, reinforced by geochemical studies and direct dating of the skeleton, now places the Liujiang fossils into an age range of ~33 to 23 ka, overturning earlier age estimates and paleoanthropological interpretations for an early modern human occupation of southern China based on this individual.

The new age estimates for the Liujiang skeletal remains are consistent with the less robust morphological characteristics compared to the chronologically older counterparts from Zhoukoudian Upper Cave[31]. Liujiang now joins well-known MIS 3/2 fossils of *H. sapiens* at Tianyuan[10], Zhoukoudian Upper Cave[11], Bailiandong[47] and Laoyadong[48]. Populations of *H. sapiens* were clearly present from north to south China during this period, ranging across latitudes extending from ~40 to 24° N, and in different ecosystems ranging over a distance of ~1800 km. The new dating information indicates that the Liujiang remains are from a major dispersal event of modern humans in Eurasia around 40,000 to 30,000 years ago. After the withdrawal of Liujiang from the pool of early migrants in East Asia, additional fossils and dates are needed to confirm the accuracy of earlier migration events in the region.

The revised age estimates for Liujiang now correspond with the well-known human fossil at Cro-Magnon in France (33-31 ka)[37]. In addition to chronological contemporaneity, the close affinity of cranial shape between Liujiang and Cro-Magnon 1[32] implies little

morphological differentiation among modern human populations in Europe and Eastern Asia, or rapid dispersal events of early modern humans across Eurasia continent around ~30 ka. Modelling of genetic data, however, indicates that there were multiple waves of introgression of archaic populations with modern humans across Eurasia, with a late wave of Denisovan-like introgression in East Asia between ~48–37 ka (contributing 0.04–0.07% to the gene pool), and a late Neanderthal-like introgression in East Asia between ~37–33 ka (contributing to 0.76–1.04% of the gene pool), at a time when the ancestral populations of Asians and Europeans had already genetically diverged from each other[49]. The Liujiang fossils were previously considered as a representative of forming regional populations[17], with evidence of larger and more protruding zygomatic bones, wide and low nasal bones, shoveling of the incisors, and congenital absence of the third molar. However, although the nasal bone is relatively flat, the nasal aperture is not as high, and the forward orientation of the anterolateral surfaces of the frontal process of zygomatic is not as pronounced as in people from the Neolithic period[30]. The cranial measurements of Liujiang are out of the range of modern variation[30]. Liujiang fossils are morphologically close to the contemporaneous Eurasian early modern humans. The presence of the features such as long cranium, rectangular orbit, short face, and occipital bunning suggests that Liujiang fossils probably represent an undifferentiated population that migrated and dispersed across Eurasia before the Last Glacial Maximum

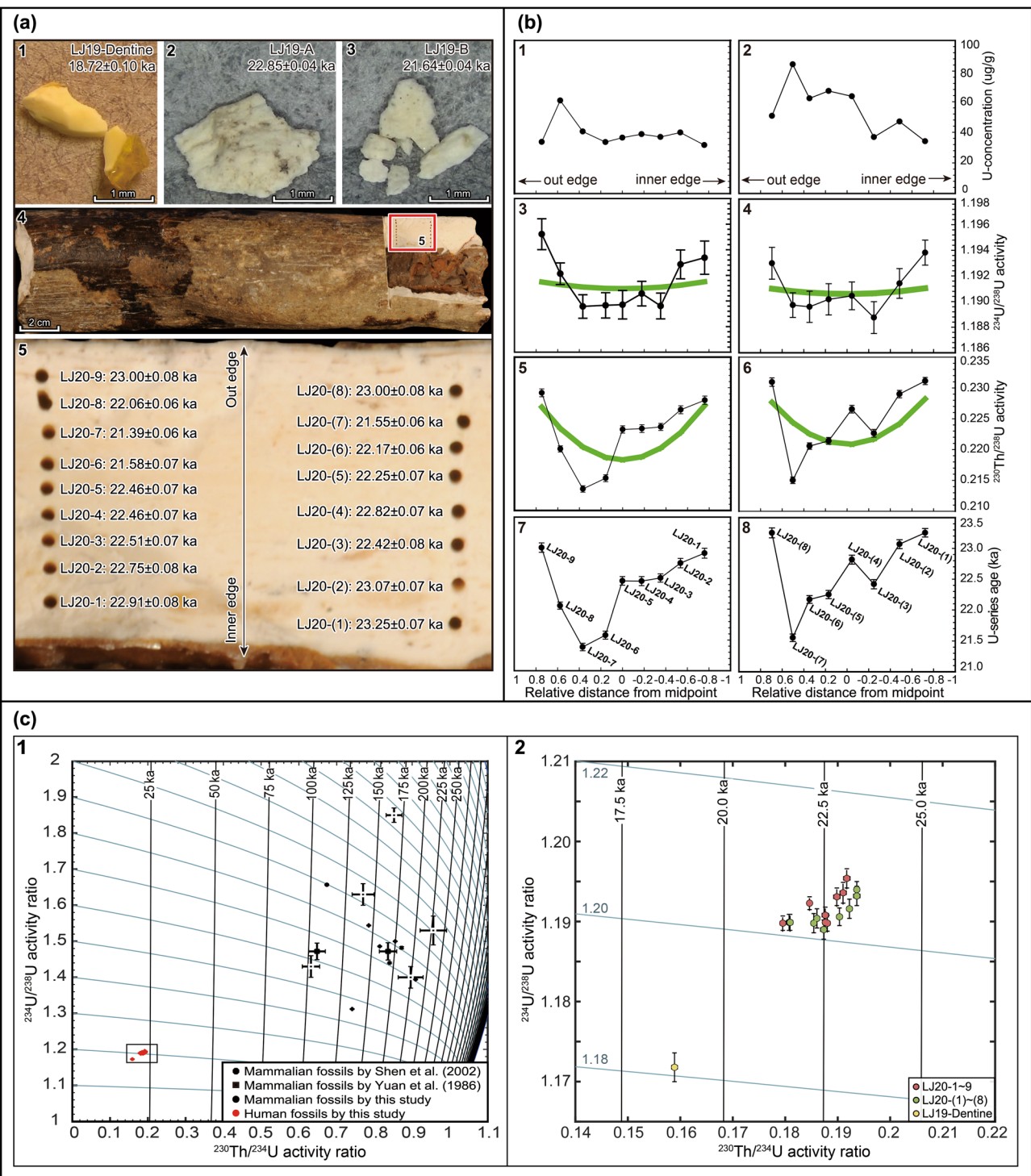

**Fig. 5 | U-series dating results on the Liujiang human fossils. a**-1, dentine fragments dropped from the upper central incisor. a-2 and -3, bone splinters from the left femur. a-4 and −5, Sampling locations showing the two bone profiles for U-series dating and associated U-series ages. The details of the red box in panel a-4 are shown in panel a-5. **b**-1−8, Distributions of U-concentration, U-series isotopic ratios and apparent U-series ages for the two sampling profiles on the Liujiang left femur consisted of subsamples LJ20-1 to −9 and subsamples LJ20-(1) to -(8), respectively. Green curves represent DAD modelling results. The values of −1.0 and 1.0 on the x-axes (b-1–8) represent the inner and the out bone edges, respectively. **c** U-series evolution diagram showing the activity ratios observed on the Liujiang dentine and femur fossils and unearthed mammalian fossils. The details of the small box in panel c-1 are shown in panel c-2. Black lines are isochrones. Blue lines show U-series evolution in a closed-system for selected initial $^{234}U/^{238}U$ values. The error bars in panels (**b**) and (**c**) represent 2σ analytical uncertainties for the data of present study and 1σ for previous studies (c-1). Source Data for Figs. 2 and 4 can be found in Supplementary Data 2.

(LGM). Thus, with the chronological data presented here, the Liujiang fossils suggest a later appearance of typical regional characteristics in Eastern Asia, probably after 23 ka. However, more pre-LGM *H. sapiens* fossils are required to further illustrate this scenario.

Tongtianyan cave is in the tropical/subtropical zone of southern China, and our dating of the Liujiang skeleton to ~33–23 ka places it at the interface of MIS 3/2 when global cooling led to forest break-up and an increase in mosaic ecosystems in the lower latitudes. Dennell and

colleagues[50] made a distinction between the northern Palearctic and southern Oriental biogeographic zones of Eastern Asia, marking differences in population movements, biological and cultural interconnections and adaptive behaviours. In northern China, new stone tool-making technology, including blades and microblades, emerges alongside bone tools suggesting rapid and widespread population migration and technological diffusion in an open steppe environment during this period[51,52]. In contrast, in southern China, microblade technology only appears around 15 ka in Niupo cave of Guizhou Province[53], with archaeological evidence mostly supporting regional adaptations to an array of ecosystems. For instance, in the subtropical Yunnan Province in southwestern China (Fig. 1a), cultural remains were dominated by cobble tools and Hoabinhian tools thought to be adaptations to forested landscapes, though some sites contain small stone tools and bone tools representing the hunting of small fauna[54,55]. In Guizhou Province (Fig. 1a), north of Liujiang, conditions were cooler and drier, and small flake tools of the core-flake tradition dominate alongside ubiquitous bone tools[55,56]. In Guangxi Province, where Liujiang is located, lithic assemblages are comprised of a mix of small tools of the core-flake tradition and cobble-tools[55]. In the contemporaneous and neighbouring sites of Bailiandong and Liyuzui (Fig. 1a), small flint tools and bone tools were identified, including forms typed as endscrapers, arrowheads and small points, thought to show similarities with Upper Palaeolithic assemblages to the north[47,55,57,58]. Highly diversified technological industries in southern China, and increased use of innovative small flint tools in contemporaneous sites near Liujiang, may be tied to novel adaptations to mosaic ecosystems in MIS 3/2 or possibly cultural influences from the north.

Modern human fossils have been reported from over fifty sites widely distributed across the East Asian mainland. However, many human fossils have unclear provenience and questionable chronological ages, as was the case with the Liujiang skeletal remains. This undermines the value of fossils for understanding modern human origins and dispersals. The present study provides a robust age range for the Liujiang skeletal remains from both a stratigraphic and chronological perspective, emphasizing the necessity to establish proper proveniences for human remains across Eastern Asia.

## Methods

Sample extraction from the Liujiang human fossils, and from the stratigraphy of the Tongtianyan cave has been permitted by the Institute of Vertebrate Paleontology and Paleoanthropology, Chinese Academy of Sciences, and Liuzhou Lotus Cave Science Museum. Results of this study will be shared with the Liuzhou Lotus Cave Science Museum, following the strategy for information sharing that we established prior to starting fieldwork in 2019.

### Radiocarbon dating

A single piece of charcoal and two organic sediment samples from Layers 1 and 2 of Unit III were collected and submitted to the BETA Analytic laboratory (Beta-) and Peking University (BA-) for radiocarbon dating. These samples were pretreated using routine acid-alkali-acid and acid-washing methods, separately. In addition, a bone sample of 402.2 mg (R-EVA 382) cut from the Liujiang femur was also pretreated at the MPI-EVA, Department of Human Evolution, Leipzig, following the method for collagen extraction and purification, including an ultrafiltration step, as described in Talamo and Richards[59]. However, the collagen retrieval failed due to its poor preservation (Sahra Talamo and Qiaomei Fu, personal communication).

The dates were reported in radiocarbon years BP (Before Present - AD 1950) using the half-life of 5568 years (in Supplementary Table 2). Isotopic fractionation was corrected for using the $\delta^{13}C$ values measured on the AMS. The quoted $\delta^{13}C$ values were measured independently on a stable isotope mass spectrometer (to ±0.3 per mil relative

to VPDB). The new $^{14}C$ determinations were calibrated using the INTCAL20 calibration curve[60] and the OxCal 4.4 platform[61,62], with age ranges expressed at the 95.4% confidence interval. All other previously published radiocarbon dating data cited in this work have also been recalibrated using the INTCAL20 calibration curve.

### U-series dating

The U-series dating analyses were carried out at the laboratory of Nanjing Normal University. For bones and teeth, the samples were weighed and dissolved with 3 N $HNO_3$ in Teflon beakers. A quantity of a $^{229}Th$-$^{233}U$-$^{236}U$ triple spike was then added into sample solutions. The sample-spike mixture was heated overnight on a hot plate at 120 °C. After the equilibration of the sample-spike mixture, U and Th were separated from each other and from other cations by passing the sample solution through a U-TEVA resin column following the procedure of Douville et al. [63]. Firstly, the sample matrix elements were eliminated through rinsing with 3 N $HNO_3$. Subsequently, Th was eluted using 3 N HCl, and finally, U was eluted using 0.5 N HCl. One drop of $HClO_4$ was added to the U or Th fractions to remove any organic material derived from the U-TEVA resins. The U and Th solutions were evaporated to dryness and then dissolved in a mixture of 0.5 N $HNO_3$ and 0.01 N HF for U and Th isotopic analyses. The protocol used for the flowstone samples was similar, with the U-Th co-precipitation with iron hydroxide prior to passing the U-TEVA resin column.

The U and Th isotopic measurements were performed on a Thermo Fisher Neptune MC-ICPMS (multi-collectors inductively-coupled plasma mass spectrometer). It is equipped with nine Faraday cups and a secondary electron multiplier (SEM). A retarding potential quadrupole (RPQ) energy filter was positioned in front of the SEM. An Aridus-II desolvator system (Cetac) coupled with an ESI-50 nebulizer and an Auto-Sampler (ASX-520) were used for sample introduction. The U-Th data acquisition strategies applied here were similar to those described by Shao et al. [64]. The U isotopic data were acquired in two static sequences. The first sequence measured $^{233}U$, $^{235}U$, $^{236}U$ and $^{238}U$ in cups and simultaneously $^{234}U$ on the SEM (with RPQ-on). The second sequence shifted all masses by 1 amu to the lower mass, so that $^{233}U$ was measured on by the SEM and the other isotopes were by the cups. Thorium measurements were carried out immediately after the uranium measurements for the same sample. $^{229}Th$ and $^{230}Th$ were measured alternately on the SEM (with RPQ-on) and $^{232}Th$ in a cup. The U isotopes of the Harwell uraninite HU-1 standard solution, which is widely used standard at secular equilibrium state, were measured after every 3 samples to monitor external reproducibility.

The amplifier gains, dark noise, hydride interferences and machine abundance sensitivity were evaluated every day prior to the sample measurements. The base lines were automatically calibrated before each U isotopic measurement. Instrument memory was assessed with the SEM by introducing a blank solution before measurements of either U or Th were conducted. The relative yields of the SEM/Faraday cups were determined during U isotopic measurements by alternating the $^{233}U$ beam (-5 mV) on the SEM and in the Faraday cup. Instrumental mass fractionation was corrected by using an exponential function by comparing the measured $^{238}U/^{235}U$ with the natural value of 137.82 for unknown samples[65]. Procedural blank was corrected by using the long-term observed values of -0.8 fg $^{234}U$, -10 pg $^{238}U$, -0.1 fg $^{230}Th$ and -2.0 pg $^{232}Th$, respectively. The U-series ages were calculated by Monte-Carlo simulations[64], using half-lives of 75,584 a for $^{230}Th$ and 245,620 a for $^{234}U$[66], $1.4 \times 10^{10}$ a for $^{232}Th$[67], and $4.47 \times 10^9$ a for $^{238}U$[68]. The fossil samples and the carbonate samples from the capping flowstone were corrected with the assumption of an initial $^{230}Th/^{232}Th$ activity ratio of 0.8 ± 0.4, which is a value for a material at secular equilibrium with the bulk Earth upper crustal $^{232}Th/^{238}U$ atomic ratio of ~3.8[69].

## Luminescence dating

The optically stimulated luminescence dating technique determines the time elapsed since the last sunlight exposure of a deposit, i.e., its burial time, which was first introduced by Huntley and colleagues[70] and has been widely used to date geological and archaeological deposits[71–74] in recent decades. The luminescence emitted from minerals (e.g., quartz and feldspar) under artificial light exposure is proportional to the absorbed energy accumulated within the crystal lattice of minerals by ionizing radiation (e.g., alpha, beta or gamma radiation) from radioactive elements such as uranium (U), thorium (Th), Rubidium (Rb) and potassium (K) in the environment, as well as cosmic rays[73,75]. By comparing natural luminescence signals with signals generated after known laboratory irradiation doses, the total radiation dose absorbed by mineral grains over the burial time (or equivalent dose, $D_e$) is determined. The assessment of the natural environmental irradiation dose rate, to which the sample is exposed during its burial history, involves measuring the radioactivity of the sample and its surroundings using chemical and radiometric methods, and estimating the radiation contributed by cosmic rays. The luminescence age of sediments is then achieved by dividing the equivalent dose (Gy) by the dose rate (Gy/ka)[76].

To provide further age constraints on the Tongtianyan sequence, eleven sediment samples (Supplementary Fig. 1a) were collected from Units II and III of the Liujiang sequence for OSL dating by hammering steel tubes (20 cm-long cylinders with a diameter of 5 cm) into a freshly dug vertical section. The tubes were then covered and sealed with aluminum foil and wrapped in black plastic bags and taped to avoid light exposure and moisture loss. The sediments from the potential light-exposed end of the cylinder were removed and those from the middle of the cylinder was used for $D_e$ measurement.

As suggested by the grain-size analyzes, coarse-grain fraction (>63 μm) was scarce in most of these samples, fine-grain (4–11 μm) quartz OSL dating was applied for them in this study. The raw samples were first etched with 10% hydrochloric acid and subsequently with 30% hydrogen peroxide to remove carbonates and organic matter. Subsequently, the chemically treated sediments were then suspended in a column of 0.01 N sodium oxalate to disperse for 20 min according to Stokes Law to remove the >11 mm fraction. This procedure was then repeated for longer 4 h periods to isolate the desired 4–11 μm poly-mineral fraction. To extract pure fine-grain quartz grains for OSL dating, these fine silt fractions were mostly etched using the 30% $H_2SiF_6$ (hydrofluorosilicic acid) for 4 days, except for samples HED-628 and HED-683, etched only three days to reduce the risk of loss of quartz due to its small sample size. The purity of the quartz extracts was confirmed by the absence of a significant infrared stimulated luminescence (IRSL) response at 60 °C to a large regenerative β-dose. Dispensing 2 mg of this fraction (4–11 μm quartz), it then suspended in small tubes with distilled water, and then was settled on each 10 mm diameter stainless steel disc for luminescence measurements. Sample preparation was carried out in the IVPP laboratories.

Luminescence measurements were made on a Risø Model DA-20 TL/OSL reader equipped with a $^{90}Sr/^{90}Y$ beta source for irradiation[77] and an EMI 9235QA photomultiplier tube. Blue light LED (470 ± 30 nm) stimulation set at 90% of 50 mW cm$^{-2}$ full power and 7.5 mm Hoya U −340 filters (transmission between 290 and 370 nm) were used for the quartz OSL measurements. The beta sources of the readers were calibrated using standard Risø calibration quartz (RCQ)[78,79] with a batch number of 108 and a correction of ~8.25% was performed[80]. Source calibrations were carried out using 4–11 μm fine-grained quartz grains on stainless steel discs.

The single-aliquot regenerative-dose (SAR) protocol was used to determine the $D_e$ value[81–83] (Supplementary Table 3). A test dose of 30 Gy, 20–30 percent of natural dose was used for $D_e$ determination. After the standard SAR protocol, a repeated regenerative dose measurement was applied to determine the recycling ratio. In addition, to test the purity of the quartz extracts, an additional recycling step was given at last to each aliquot, in which IR stimulation of grains for 40 s at room temperature using infrared LEDs was set after preheat and prior to blue light stimulation. The OSL signals of the first 0.8 s (channels 1–5) integral after late background subtraction from the last 4 s (channels 225–250) were used for dose response curve construction[84]. These measured aliquots which having the recycling ratio or OSL IR depletion ratio exceeding the acceptable range (0.9–1.1), or with the recuperation over 5%, were excluded from the $D_e$ determination[83,85]. For each sample, there were only ~1–2 aliquots (0.5–0.7%) that failed to meet the above criteria, and these were rejected for $D_e$ determination. Finally, 14–40 aliquots were accepted for $D_e$ determination of each sample, and the arithmetic mean was used for $D_e$ calculation of each sample.

Prior to the routine application of the SAR protocol, several laboratory tests were carried out on the extracted quartz grains. Firstly, to select an appropriate preheat temperature for $D_e$ determination, preheat plateau tests for two representative samples (HED-640 from Layer 2 and HED-682 from Layer 4) were conducted. The preheat temperature varied from 160 °C to 300 °C at 20 °C increments, with the cut-heat ranged from 160 °C to 260 °C at 20 °C increments basically tracking and lagging the preheat temperatures by a margin of 40 °C[86]; for each preheat temperature step, the mean $D_e$ value of three aliquots was calculated. To confirm the suitability of measurement conditions, a dose recovery test for fine-grain quartz using the SAR protocol outlined in Supplementary Table 3 was also carried out on 8 samples with given doses ranging between 110.2–142.6 Gy. For the other three samples including HED-626, HED-638 and HED-639 from the Unit-II with the quartz completely saturated, dose recovery tests were not conducted. At least three aliquots for each sample were first bleached twice at room temperature for 100 s with blue LEDs, with a pause of >5000 s in between to avoid any charge transferred to the 110 °C thermoluminescence (TL) trap. A known beta dose approximating the natural dose was then administered to each aliquot, and the same approach as the $D_e$ measurement was used to recover this known dose. The dose recovery ratio was then determined by dividing the measured $D_e$ by the given dose (Supplementary Fig. 9). Finally, a preheat temperature of 260 °C and a cut-heat at 220 °C were used for De determination of all samples.

To determine the environmental dose (from natural $^{238}U$, $^{232}Th$, $^{226}Ra$ and their decay products, and $^{40}K$) of these OSL samples, sediments from a 30 cm radius around the sampling tubes were collected for dosimetry measurements and water content. The specific activities of radionuclides including $^{238}U$, $^{226}Ra$, $^{232}Th$, $^{210}Pb$ and $^{40}K$ were mainly measured using high-resolution gamma γ-spectrometry (HRGS)[87]. Firstly, the bulk samples were dried at 60 °C and ground for homogenization. Then, about 280–300 g dry powder sediments for each sample were sealed into a plastic container and stored for 1 month to build up equilibrium between $^{222}Rn$ and $^{226}Ra$ before determining the activities of natural U, $^{232}Th$ and their daughter nuclides and $^{40}K$. All samples were measured for 24 h using a high-purity germanium detector (ORTEC GEM70P4-95, P-type, 70% relative efficiency, 122 kev FWHM at 1.0 kev and 1.332 Mev FWHM at 2.0 kev) shielding in low-activity lead to minimize the influence of environmental radioactivity. In addition, a field γ-spectrometer with a NaI detector was also used to estimate the in-situ gamma dose rate at the sample location to reduce the effect of heterogeneous sediments and the effect of Rn escape. Each sample was measured for 8 h in the field. Spectral data were converted to activity concentrations and infinite matrix dose rates using conversion data by Guérin et al. [88]. The cosmic dose contribution was estimated by taking account of the burial depth of the sample, the thickness of the cave roof overhead (12.5 m), the zenith-angle dependence of cosmic rays, and the latitude, longitude, and altitude of the site[89]. Water content (moist mass/dry mass) was determined by weighing the sample before and after drying and was assigned an

absolute uncertainty of ±3%. Subsequently, using the dose rate conversion factors of Guérin et al. [88] and water content attenuation factors[76], the radioactive element concentrations were converted into an effective dose rate. The alpha dose was calculated using an alpha efficiency value of $0.04 \pm 0.02$ according to Rees-Jones[90]. The beta grain size attenuation factors from Guerin et al. [88] were used for beta dose rate calculation. Finally, dose rates and ages were calculated with the ager program[91].

## Bayesian age modelling of the Tongtianyan cave deposits

To establish a chronological framework for the depositional units at Tongtianyan cave, we conducted Bayesian analysis that includes the radiocarbon, OSL and carbonate U-Th dating data, using OxCal v4.4[61,62]. All $^{14}$C ages were calibrated using the INTCAL 20 dataset. For all the OSL and U-series ages, we used N_date in calendar years before AD 2022 with associated 1σ errors as likelihood estimates. Additionally, the U-series dates for the Liujiang human fossils were also included in the model as minimum age estimates using a *Before* command. To estimate the posterior distributions (i.e., the modelled ages), the stratigraphic order of each sample was input using the Sequence function (Supplementary Code 1), based on the assumption that a sample stratigraphically lower is older than those above. For samples that are from the same stratigraphic layer with similar depth or with ambiguous stratigraphic orders, such as Layers 3, 2 and 1, they were modelled as a *Phase*, in which the measured ages are assumed to be part of a single phase. For the layers with abrupt and clear boundaries between them, such as between Layers 3 and 2, or Layers 2 and 1, upper or lower Boundaries were placed for them to constrain their start or end ages. Otherwise, for layers for which gradual stratigraphic changes were identified, assuming continuous sediment accumulation, transitional boundaries were placed between them. The samples, phases and sequences were arranged according to their relative stratigraphic order.

We applied the general t-type outlier model[92] to detect outlier ages by assessing the likelihood of each age being consistent with the modelled ages. A prior outlier probability of 5% was assigned for the other samples with their posterior outlier probability calculated during the modelling process. The codes used to run the Bayesian model are listed in Supplementary Code 1. The generated probability distribution functions (PDF) for each of the samples are shown in Fig. 4b and their corresponding 95.4% probability ranges are summarized in Supplementary Data 5. As demonstrated in Fig. 4b, the Bayesian model results in improvements in the precision in age estimates for all the samples. None of the samples was flagged as an outlier, as indicated by the posterior outlier probabilities, which are less than 4% for all samples. The model is shown in Supplementary Information section 6.2.

## Sr-Nd isotopic analysis

Five representative samples were analyzed for Nd-Sr isotopic and trace element analysis, including the samples of LJHS and one each from Layer 2 (HED-628), Layer 3 (HED-627) and Layer 5 (HED-683), as well one from the Unit II (HED-626). The silicate Nd-Sr isotopic ratios of the samples were determined by a Finnigan Neptune MC-ICP-MS at the Nanjing University following the method of Chen et al. [93]. About 1 g of powdered sample was treated with purified acetic acid solution (0.5 mol/L) at room temperature for up to 8 h to remove the carbonate and then heated at ~600 °C for 8 h to remove organic matter. Subsequently, ~0.1 g of this treated sample was transferred to Savillex vials to digested with acids (2 mL of HNO₃ + 1 mL of HF) at 100 °C for 24 h and dried. About 3 mL of aqua regia was added to the dried sample and sealed vials were heated at 110 °C for ~12 h. After addition of 3 mL of 2 N HCl in steps, the vials were kept at 110 °C for drying, following which the samples were ready for ion exchange column chemistry. Standard ion exchange protocols were followed for the separation of pure Sr and Nd fractions. The $^{87}$Sr/$^{86}$Sr and $^{143}$Nd/$^{144}$Nd ratios were then measured on a MC-ICP-MS. The analytical blanks are insignificant: <1 ng for Sr and <50 pg for Nd, respectively. Reproducibility and accuracy were checked by running the strontium standard SRM NBS 987 and neodymium standard La Jolla, with mean $^{87}$Sr/$^{86}$Sr of $0.710264 \pm 0.000013$ (external ± σ, $n = 10$) and mean $^{143}$Nd/$^{144}$Nd of $0.512121 \pm 0.000007$ (external ± σ, $n = 10$) respectively.

## Trace element analysis

Five samples for trace element analysis were leached of calcium carbonate using 1 mol/l acetic acid (HAc) to totally leach the carbonate fraction without significant effect on silicates or iron oxides. Before acid dissolution, all samples were finely ground using an agate mortar. Then, 40 mg these samples were dissolved in Teflon bombs with a stainless steel jacket with 0.3 mL HNO₃ and 1 mL HF added. After being shaken with ultra-sonic device for 10 min, the bombs were placed on a hot plate (170 °C) for 15 h. The solutions were then dried, and a HNO₃-HF solution was added. The bombs were again placed on a hot plate (20 °C) for 7 days. After the solutions were dried, 3 mL HNO₃ was added, and the bombs were placed on a hot plate (170 °C) for 5 h. The solutions were then evaporated, dissolved in 3 mL HNO₃ and placed on a hot plate for 5 h. The solutions, after cooled, were then transferred into 50-mL volumetric flasks, and 1 mL 500 mg/L Indium solution was added as the internal standard. The final solutions were diluted by 1% HNO₃. The trace element compositions were determined using an ICP-MS (ELEMENT, Finnigan MAT) in the Nanjing University. The rhodium solution (10 ppb) was dropped into the sample solutions for instrument drift correction during the measurements. The analytical uncertainties are less than 5% for each element.

## Grain size analysis

Nineteen sediment samples including eleven OSL samples and eight bulk samples from the three sedimentary units on the Liujiang section, and one bulk sample from the sediments trapped in the cavity of the Liujiang femur, were collected for grain-size analysis (Supplementary Fig. 1). These samples were firstly pretreated with 30% hydrogen peroxide (H₂O₂) to remove organic matter, and then treated with 10% hydrochloric acid (HCl) to remove carbonates. Subsequently, the samples were dispersed with 0.5 N sodium hexametaphosphate ((NaPO₃)₆). Grain-size was determined using a Malvern Mastersizer 3000 particle analyzer with an analytical precision of <1%. The Mie theory was applied with a particle refractive index of 1.520.

## Colour analysis

All these nineteen sediment samples were dried at 50 °C and ground to a powder for colour analysis. The measurements were conducted using a Courtney Minolta CM-400 colorimeter under standardized observation conditions (2° Standard Observer, Illuminant C), with a D65 standard light source. The spectral information was converted into the CIELAB Colour Space system (CIE 1976). The L × a × b values indicate extinction on a scale from L × 0 to L × 100, and express colour as chromaticity coordinates on red-green (a*) and blue-yellow (b*) scales.

## Reporting summary

Further information on research design is available in the Nature Portfolio Reporting Summary linked to this article.

# Data availability

All relevant data are available in the main text or the accompanying supplementary materials. Source data are provided as a Source Data file. The Liujiang human and mammalian fossils referred to in this study are currently in the Institute of Vertebrate Paleontology and paleoanthropology, Chinese Academy of Sciences, Beijing, and at the Nanjing Normal University in Nanjing, China. Source data are provided with this paper.

## Code availability

All relevant code is available as a Supplementary Code file.

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

## Acknowledgements

We thank Wu Liu, Xiujie Wu and Qiaomei Fu (Institute of Vertebrate Paleontology and Paleoanthropology, Chinese Academy of Sciences), Chris Stringer (Department of Earth Sciences, Natural History Museum, London, UK), Sahra Talamo (MPI-EVA, Department of Human Evolution, Leipzig, Germany), Andrew S. Murray (Aarhus University, Aarhus, Denmark) and Jun Peng (Hunan University of Science and Technology, Hunan, China) for discussions; Yan Li and Chi Ma (Lotus Cave Science Museum, Liuzhou) for assistance with field sampling; Faxiang Huan, Jiequn Hua and Ruiping Tang (Institute of Vertebrate Paleontology and Paleoanthropology, Chinese Academy of Sciences) for assistance with

figure preparation and OSL sample preparation; Yan Li (China University of Geosciences, Beijing, China) and Gaojun Li (Nanjing University, Nanjing, China) for helpful discussions on optical dating and geochemical analyses. Financial support for this research was provided by the NSFC Basic Science Center Program (41888101 to Z.T.G., C.L.D. and J.Y.G.), the Chinese Academy of Sciences (YSBR-019 to S.X.), NSFC grants 41977380 to J.Y.G, 41877430 to Q.F.S., and 42372011 to S.X., the Strategic Priority Research Program of the Chinese Academy of Sciences (XDB26000000 to S.X., J.Y.G. and X.G.), and Griffith University funding to M.P.

## Author contributions

J.Y.G., S.X., Q.F.S., Z.T.G. and C.L.D. obtained funding and initiated the project; Q.F.S., J.Y.G., S.X. and X.G. coordinated the research. J.Y.G., S.X., Y.J.J. and Q.F.S. conducted fossil and site sampling; J.Y.G., Q.F.S, and C.L.D. conducted stratigraphic and provenancing studies; J.Y.G. and H.L.Y. performed OSL dating; Q.F.S., and T.Y.J. performed U-series dating; X.S. analyzed comparative studies of the Liujiang human fossils; and J.Y.G., S.X., Q.F.S, M.P, R.G., S.X.Y., and K.L.Z. wrote the main text and supplementary materials with specialist contributions from the other authors.

## Competing interests

The authors declare no competing interests.
