## [Peer Review File · Nature Communications]

New Late Pleistocene Age for the Homo sapiens skeleton from Liujiang southern ChinaEditorial Note: This manuscript has been previously reviewed at another journal that is not operating a transparent peer review scheme. This document only contains reviewer comments and rebuttal letters for versions considered at Nature Communications.

Reviewers' Comments:

Reviewer #2:

Remarks to the Author:

New Late Pleistocene Age for the Homo sapiens skeleton from Liujiang, southern China

The new dating of the skeletal remains from Liujiang cave is a useful addition to our knowledge of the history of H. sapiens in China. The account of the dating is detailed, thorough and convincing, and should end any controversy over the age of this material. The description of the skeletal evidence is also useful and provides a more up-to-date overview of its coverage in the Wu and Poirier 1995 volume Human Evolution in China (Oxford 1995).

Regarding publication in Nature, my main concern is that Liujiang has now lost its international significance. Its new dating at 33-24 ka brings the skeleton in line with other East Asian finds, such as Salkhit, (Mongolia), Upper Cave, Zhoukoudian, Moh Thiew (Thailand); and shows that it is younger than the specimens from Tianyuandong, North China, Niah Cave, Borneo and Tam Pa Ling, Laos, all of which are >40 Ka. As a result of the re-dating, Liujiang is now in the category of "interesting but not very important" in international terms. I therefore think it is inappropriate for Nature. There is no doubt that this is an excellent paper; however, it is one that has a regional but not international significance.

The redating and skeletal descriptions are certainly worth publication in a journal such as Journal of Human Evolution, where the new evidence can be read by the relatively small community familiar with the human skeletal record of China and SE Asia.

Reviewer #5:

Remarks to the Author:

Dear Authors,

Thank you for the opportunity to read your manuscript NCOMMS-23-34915-T "New Late Pleistocene Age for the Homo sapiens skeleton from Liujiang, southern China". I have been invited to share my thoughts on a broader level for the manuscript, which I am doing in the following, I will not comment much on individual items raised by your four other excellent reviewers. I am glad to see that most of their suggestions have been integrated into the new paper.

I agree with all previous reviewers and would like to add my own voice to how important the contribution of this paper is towards the debate and research on the dispersal of Homo, particularly Homo sapiens into Asia, I think it will provide a great service to the whole palaeoanthropology and Human Evolution community and move the field forward and for that alone, I would love to see the paper published.

I would also like to say that I think that the comments and suggestions made by the previous reviewers were all valuable and in support of making this a concise and high-quality manuscript. I would therefore urge the authors to make sure that the suggestions are worked into the new version

of the manuscript. Overall, this is well done, I would however ask if you could consider adding the Unit I label to current figure 3 in the main text as reviewer #3 requested to improve the figure. I checked your suggestions of the new figure in the supplementary material and the label is missing there too. I need to insist on adding this. There are a couple of other points I would like to stress further but I hope it is not adding too much extra work.

- The use of the term provenance vs. providence

I am a bit concerned about the use of the term provenance – it is sometimes not clear to me if it is used correctly in the paleoanthropological sense of it. Provenance in palaeoanthropology is used to describe the “history” of a fossil/artefact since its discovery – e.g., the lost fossils from Zhoukoudian have a very interesting provenance from the time they were discovered until their loss, the same can be said about e.g., the Le Moustier 1 Neanderthal, which was dragged all over Europe and damaged during the second World war.

On the other hand, provenience is used to describe the exact location or place of discovery of an artifact (artifacts can also be soil samples) or a fossil. Given that this is a new dating paper, which provides a lot of information about identifying the relevant sediment layers, it might be more appropriate and scientifically correct to talk about provenience rather than provenance in the paper.

- Overall flow of the paper

The different parts of the paper read well but overall; the flow of the manuscript appears now a bit disconnected. Perhaps this is to do with the intense review process already undertaken. Some of the sections do not follow well on each and changes in topic can be very abrupt. I would think the use of subtitles for different sections could help with that. I would also suggest a reorganisation of some of the sections to increase the flow.

Most importantly, the order of the figures is currently not coherent and not following convention. I have suggested of reorganization of the three first figures and some of the sections of the main text in relation to figures 4 and 5 to bring more flow into the story and to pull it all together (again). The current version of the section titles is of course not mandatory in this form, feel free to adjust them if you wish.

I will provide these suggestions a track-changes version of the manuscript with my own additions, they are also refining some of the English and add some more precision to the anatomical description of the skeletal remains.

Generally, I think the added details after the revisions for the skeletal remains is already good, it clearly states that the remains can be attributed to modern *Homo sapiens*, I have just added more clarification to the use of anatomical terms. I have – based on the requests from several reviewers and your arguments following this - made a couple of changes where you use the term “skeleton” with e.g., remains.

This is because I agree with the argument that the specimen is indeed only a partial skeleton but also I can see why you would argue for the “most complete” claim as there are indeed not that many postcranial elements for Asian fossils. Where I made the suggestions, I am just toning down the term most complete/skeleton where it is not related to the argument that this is the most complete set of remains in East Asia/Southern China. I am glad that the authors also have highlighted the combination of cranial characteristics with associated postcranial remains.

I will also provide a cleaned-up version of the manuscript with all the changes accepted to make the new suggested shape of it more visible and keeping track all the changes suggested.

I would be very grateful if you could give these suggestions your consideration,

Kind regards,

Sandra Martelli

Reviewer #6:

Remarks to the Author:

Modern human skeleton fossil from the Tongtianyan Cave, Liujiang is one of the most complete *H. sapiens* skeleton fossil findings in southern China. The chronology of human fossil from this site is important for understanding the dispersal history of modern humans in East Asia. Previous chronological works on this site suggested an early arrival of modern human in East Asia, but these ages were controversial. This study establishes new chronologic framework for this key site using OSL, U-series and radiocarbon dating, and suggest a younger age for human fossil at Liujiang in correspondence with ages of other human fossils in northern China. The human fossil at Liujiang itself was dated and its provenance in the stratigraphy of the cave is assessed using a comprehensive approach. The paper is well-written and scientifically sound; experiments and results are robust and are clearly present. I think the findings of this paper will attract broad interest for readers of *Nature Communication*. An earlier version of the manuscript has been submitted to *Nature* and the authors have properly addressed the comments of four reviewers. Therefore, I think this manuscript is already of high quality, and recommend acceptance of this paper after minor revision. Some minor comments are listed below.

Main text:

Lines 70-71: The chronologies of modern human remains from the Tianyuan Cave and the Zhoukoudian Upper Cave listed here are slightly different from those listed in the abstract. According to Shang et al., 2007, *PNAS* and Li et al., 2018, *JHE*, bone samples from human fossil-bearing layers of the Tianyuan Cave and the Zhoukoudian Upper Cave were dated to 42-39 ka cal BP and 38.3-35.8 ka cal BP, respectively. Please check these literatures and use consistent numbers in the manuscript.

Line 80: I suggest the authors cite some references here, regarding the provenance and dating of the Liujiang hominin fossils.

Lines 81-198: These paragraphs are very informative and I can get information regarding the investigation history of the Tongtianyan Cave, previous dating works of the cave, features of human fossils, geomorphologic setting and stratigraphy of the cave (and potential location of human fossils) by reading through these paragraphs. But to make them easier to follow, I suggest the authors may consider re-order these paragraphs. Personally, I would use such a structure: 1) the location and geomorphologic setting of the Tongtianyan Cave; 2) a brief investigation history of the cave; 3) the stratigraphy of the cave, archaeological findings of different stratigraphic units and potential location of human fossils; 4) features of human fossils; and 5) previous dating works and their shortage. I reiterate that these paragraphs are already adequately informative, so this structure issue is just a personal advice.

You may consider swapping Figure 2 and Figure 3, as in this manuscript Figure 3 (first referred to in Line 163) is mentioned earlier than Figure 2 (first referred to in Line 213).

Figure 3: There is no legend for Fig. 3b and Fig. 3c, and I can't see the stratigraphic drawings in Figs. 3b and 3c very clearly. The authors may consider plot Fig. 3b and Fig. 3c separately and add legends in the figures.

Lines 345-347: I can't understand this sentence. Please rephrase.

Lines 355-357: Are there any other well-dated sites on this dispersal route supporting this statement?

Methods

Line 659: Rb is another ionising irradiation source, although typically with a low content in sediments.

Line 661: this sentence may be changed to "the total radiation dose absorbed by mineral grains over the burial time..."

Line 684: which sample/samples was/were etched for a shorter time?

Line 692: Reference 85 is not about OSL reader. Please check the references and make sure they are cited properly.

Line 697: have you calibrated your reader using new batches of Riso calibration quartz (Batch 126 and onwards)? If yes, how large is the difference between the apparent machine dose rate obtained from the new batch and the 108 batch? Does it validate the usage of an 8.25% correction factor?

Lines 701-702: This sentence is unclear. Do you mean the repeated regenerative dose for recycling ratio measurement?

Lines 704-705: revise to a phrase like "...after preheat and prior to blue light stimulation" and remove repetitive words "with IR stimulation carried out prior to blue stimulation".

Lines 708-710: This sentence is not very clear, please rephrase. How long time did you use for OSL measurement?

Line 713: At the end of this paragraph please clarify how did you estimate the mean De values for your samples.

Lines 719-720: Do you mean 40°C lower than preheat temperatures?

Line 732: Finally in this paragraph, please clarify what are the final preheat and cutheat temperatures used for dating.

Line 749: I guess you mean "reported by Olley et al"? And I am confused here.... In the supplementary materials you say that the program by Murray et al., (2021) was used for dose rate calculation; in their program, the conversion factors are derived from Guerin et al. (2011) rather than Olley et al. (1996). So, which conversion factors did you use exactly for dose rate estimate? Or do you mean you applied the conversion factors of Olley et al. (1996) for field gamma spectrometry data analysis? Please clarify.

Line 760: Reference 95 is absent in the reference list of the main text, and that reference in the Supplementary materials is not related to OSL dating. Please check the cited reference.

Supplementary Materials

Lines 49-67: There are some repetitions between this paragraph and the main text. Please remove repetitive sentences.

Line 184: "Extended data Fig. 2"—I guess you mean Supplementary Fig. 2

Supplementary Fig. 2: In figures or in the caption, please indicate which layers are these samples from. For each panel, please use different colors or line styles to differentiate samples.

Line 438: It should be "In Steps 2 and 6"

Line 439: response to a regenerative dose

Line 441: remove "by"

Supplementary Table 5, step 1: the superscript of D_i should be "a,b"

Line 443: I think here you mean "to monitor the efficacy of sensitivity change correction" or "to measure the recycling ratio"

Line 461: "the preheat plateau test results..."

Line 489: I guess you excluded aliquots with recuperation values $>5\%$

Line 491-502: The authors take $2D_0$ values as the upper dating limit for their samples. This upper dating limit estimate, however, is debatable as it is empirical and is based on single saturating exponential growth curve fitting. There is no mathematic validation that D_e values greater than $2D_0$ must be inaccurate—although D_e estimates at higher doses can be less precise due to the flatter growth curve at higher dose range, reliable D_e can be obtained if the shape of the growth curve can be accurately established and if untruncated natural signals are properly used for D_e estimate (e.g. Li et al., 2017). If the growth curves of the samples have an additional linear or exponential component, there is a potential to obtain accurate D_e estimates at higher doses (e.g. Murray et al., 2008; Fu et al., 2017), and for these growth curves the $2D_0$ estimate is inaccurate. The authors use $2D_0$ values to represent the minimum ages for three samples from Unit II. I suggest you use the real measured D_e values to estimate the minimum ages for these samples. Even though these D_e values can be imprecise, these real measured D_e values may not necessarily be wrong (but you can prudently treat them as minimum D_e values), and they can at least provide a more useful lower bound on the true D_e value than the $2D_0$ value (see discussions of Galbraith and Roberts, 2012).

Line 491: $2D_0$ represents 86% of the saturation intensity

Line 491: "monomolecular fit"—I guess you mean "single saturating exponential growth curve fitting"

Lines 492-493: this sentence is not very clear, please rephrase

Lines 494-495: It appears to me that your aliquots have very different D_0 values. As you used fine grains for dating, I expect the growth curves of different aliquots should be very similar due to strong average effect. Why the D_0 values are so different?

Line 499: "infinite ages"—I guess this should be "minimum ages"

Lines 503-520: Typically, the OD value and D_e distribution pattern are used for assessing the completeness of OSL signal resetting for single-grain OSL dating. But since this study uses fine grain (4-11 μm) quartz for OSL dating, the apparent OD value for each sample may not represent genuine between-grains difference in D_e value, as in each aliquot there can be tens of thousands of grains

contributing to the OSL signal so any between-grain difference in D_e values is averaged. This explains why you got small OD values and very concentrated D_e distribution for all your samples. I think for fine grain dating the OD value and D_e distribution pattern provide little useful information about signal resetting, but the $D_e(t)$ plots, as provided by the authors in their rebuttal letter to the reviewers of *Nature*, can provide evidence that the OSL signals are sufficiently bleached, and the consistency between OSL ages and radiocarbon ages also suggest good signal bleaching. The authors may consider adding the $D_e(t)$ plots (R-Fig.2) into the Supplementary Information.

Line 524-548: For samples exhibiting U-series disequilibrium, it would be better to estimate the impact of U-series disequilibrium on dose rate quantitatively, if possible.

Supplementary Fig. 13: It would be great if you could indicate which sample is from with unit/layer in this figure.

Supplementary Table 7 and Table 8: The errors of OSL ages present in Supplementary Table 7 (given as 1σ error) and Supplementary Table 8 (given as 2σ error) are the same, so I guess these errors are random-only errors (if the authors use random-only errors in the Bayesian age model). It would be great if the authors could provide random plus systematic error for each age in Supplementary Table 7, which would represent full uncertainty for each age estimate.

References

- Fu, X., Cohen, T.J., Arnold, L.J., 2017. Extending the record of lacustrine phases beyond the last interglacial for Lake Eyre in central Australia using luminescence dating. *Quaternary Science Reviews* 162, 88–110.
- Galbraith, R.F., Roberts, R.G., 2012. Statistical aspects of equivalent dose and error calculation and display in OSL dating: An overview and some recommendations. *Quaternary Geochronology* 11, 1–27.
- Guérin, G., Mercier, N., Adamiec, G., 2011. Dose rate conversion factors: update. *Ancient TL* 29 (1), 5–8.
- Li, B., Jacobs, Z., Roberts, R.G., Galbraith, R., Peng, J., 2017. Variability in quartz OSL signals caused by measurement uncertainties: Problems and solutions. *Quaternary Geochronology* 41, 11–25.
- Li, F., Bae, C.J., Ramsey, C.B., Chen, F., Gao, X., 2018. Re-dating Zhoukoudian Upper Cave, northern China and its regional significance. *Journal of Human Evolution* 121, 170–177.
- Murray, A., Arnold, L.J., Buylaert, J.-P., Guérin, G., Qin, J., Singhvi, A.K., Smedley, R., Thomsen, K.J., 2021. Optically stimulated luminescence dating using quartz. *Nat Rev Methods Primers* 1, 1–31.
- Murray, A., Buylaert, J.-P., Henriksen, M., Svendsen, J.-I., Mangerud, J., 2008. Testing the reliability of quartz OSL ages beyond the Eemian. *Radiation Measurements* 43, 776–780.
- Olley, J.M., Murray, A., Roberts, R.G., 1996. The effects of disequilibria in the uranium and thorium decay chains on burial dose rates in fluvial sediments. *Quaternary Science Reviews* 15, 751–760.
- Shang, H., Tong, H., Zhang, S., Chen, F., Trinkaus, E., 2007. An early modern human from Tianyuan Cave, Zhoukoudian, China. *Proc Natl Acad Sci* 104, 6573–6578.

Responses to Reviewers' Comments

We appreciate reviewers' constructive comments and suggestions on the previous version of our manuscript, which have been addressed point-by-point as follows.

REVIEWER COMMENTS

Referee #2:

New Late Pleistocene Age for the *Homo sapiens* skeleton from Liujiang, southern China

The new dating of the skeletal remains from Liujiang cave is a useful addition to our knowledge of the history of *H. sapiens* in China. The account of the dating is detailed, thorough and convincing, and should end any controversy over the age of this material. The description of the skeletal evidence is also useful and provides a more up-to-date overview of its coverage in the Wu and Poirier 1995 volume *Human Evolution in China* (Oxford 1995).

AR: We thank the reviewer for the positive evaluation of our manuscript. We appreciate the comment that the paper was “detailed, thorough and convincing” and we are pleased to see that the reviewer believes that our new dating program has ended the controversy surrounding the Liujiang skeletal remains. Moreover, we appreciate that the reviewer finds our skeletal descriptions to be a useful addition to the literature on important Chinese fossils.

Regarding publication in *Nature*, my main concern is that Liujiang has now lost its international significance. Its new dating at 33-24 ka brings the skeleton in line with other East Asian finds, such as Salkhit, (Mongolia), Upper Cave, Zhoukoudian, Moh Thiew (Thailand); and shows that it is younger than the specimens from Tianyuandong, North China, Niah Cave, Borneo and Tam Pa Ling, Laos, all of which are >40 Ka. As a result of the re-dating, Liujiang is now in the category of “interesting but not very important” in international terms. I therefore think it is inappropriate for *Nature*. There is no doubt that this is an excellent paper; however, it is one that has a regional but not international significance.

The re-dating and skeletal descriptions are certainly worth publication in a journal such as *Journal of Human Evolution*, where the new evidence can be read by the relatively small community familiar with the human skeletal record of China and SE Asia.

AR: We appreciate the positive statements by the reviewer about our paper and the importance of the Liujiang skeletal material. However, we respectfully disagree with the reviewer that the fossil material has “lost its international significance” and now not appropriate for *Nature*.

We would like to emphasize that Liujiang is one of the most recognized fossils of a modern human in Eastern Asia, which has been widely cited and studied. As a representative of ‘early’ modern humans in East Asia, Liujiang has gained international significance, serving as important material for understanding the dispersal of modern humans across Eurasia.

Our re-dating of the site stratigraphy and skeleton is one of the most advanced studies of its kind in China and all of Asia. The origin and dispersal of *Homo sapiens* in Asia has been the

subject of major debates, and much of the problem about the timing of migrations has been the lack of dating, including direct dating, on fossil material. We not only show that the Liujiang skeleton is young, and not representative of early *Homo sapiens* populations, but we also show scholars across the world that the methods that we apply are necessary to establish accurate dating. This has major implications for theories about human origins and dispersals.

Even though Liujiang is younger than anticipated, the reviewer is correct that it now forms a key fossil to demonstrate the wider occupation of Eastern Asia in the later Pleistocene, i.e., after ca. 40 ka. The fossil skeleton at Liujiang is a rare find and it now provides important new information on human demography in this critical time period.

Referee #5:

Dear Authors,

Thank you for the opportunity to read your manuscript NCOMMS-23-34915-T “New Late Pleistocene Age for the Homo sapiens skeleton from Liujiang, southern China. I have been invited to share my thoughts on a broader level for the manuscript, which I am doing in the following, I will not comment much on individual items raised by your four other excellent reviewers. I am glad to see that most of their suggestions have been integrated into the new paper.

AR: We thank the reviewer for pointing out that the suggestions of the reviewers has been well integrated into our revised paper.

I agree with all previous reviewers and would like to add my own voice to how important the contribution of this paper is towards the debate and research on the dispersal of Homo, particularly Homo sapiens into Asia, I think it will provide a great service to the whole palaeoanthropology and Human Evolution community and move the field forward and for that alone, I would love to see the paper published.

AR: We appreciate this comment about the importance of the Liujiang skeleton (in contradiction to the points raised by Reviewer #2).

I would also like to say that I think that the comments and suggestions made by the previous reviewers were all valuable and in support of making this a concise and high-quality manuscript. I would therefore urge the authors to make sure that the suggestions are worked into the new version of the manuscript. Overall, this is well done, I would however ask if you could consider adding the Unit I label to current figure 3 in the main text as reviewer #3 requested to improve the figure. I checked your suggestions of the new figure in the supplementary material and the label is missing there too. I need to insist on adding this. There are a couple of other points I would like to stress further but I hope it is not adding too much extra work.

AR: In our revised manuscript, Unit I was added in MAIN TEXT Fig. 2 and Supplementary Fig.1. As this sedimentary unit is beneath Unit III and Unit II, and is not exposed in this section, but only exposed on its contrary side of this section, the upper boundary of which thus is only presented in a dashed line (Fig. 2 and Supplementary Fig.1a).

• The use of the term provenance vs. providence

I am a bit concerned about the use of the term provenance – it is sometimes not clear to me if it is used correctly in the paleoanthropological sense of it. Provenance in palaeoanthropology is used to describe the “history” of a fossil/artefact since its discovery – e.g., the lost fossils from Zhoukoudian have a very interesting provenance from the time they were discovered until their loss, the same can be said about e.g., the Le Moustier 1 Neanderthal, which was dragged all over Europe and damaged during the second World war.

On the other hand, provenience is used to describe the exact location or place of discovery of

an artifact (artifacts can also be soil samples) or a fossil. Given that this is a new dating paper, which provides a lot of information about identifying the relevant sediment layers, it might be more appropriate and scientifically correct to talk about provenience rather than provenance in the paper.

AR: Thank you for your suggestion. In our revised manuscript, we replaced “provenance” with the term “provenience” (Main text Lines 48, 79, 162 ...).

- Overall flow of the paper

The different parts of the paper read well but overall; the flow of the manuscript appears now a bit disconnected. Perhaps this is to do with the intense review process already undertaken. Some of the sections do not follow well on each and changes in topic can be very abrupt. I would think the use of subtitles for different sections could help with that. I would also suggest a reorganisation of some of the sections to increase the flow.

AR: Thank you for your suggestion. In our revised manuscript, subtitles were used for each section according to your suggestion. We also rearranged some descriptions in our revised manuscript to improve the flow (Main text Lines 60, 175, 206, 243, 338).

Most importantly, the order of the figures is currently not coherent and not following convention. I have suggested of reorganization of the three first figures and some of the sections of the main text in relation to figures 4 and 5 to bring more flow into the story and to pull it all together (again). The current version of the section titles is of course not mandatory in this form, feel free to adjust them if you wish.

I will provide these suggestions a track-changes version of the manuscript with my own additions, they are also refining some of the English and add some more precision to the anatomical description of the skeletal remains.

AR: Thank you for your suggestion. To improve the flow of the paper, Figure 2 and Figure 3 have been swapped in the revised main text (which is also suggested by Referee #6), and their captions have also been exchanged. In addition, we moved Fig. 4b&c into Fig. 5, as Fig. 4b&c is mentioned after Fig. 5. The order of the figures is now coherent and follows convention.

Generally, I think the added details after the revisions for the skeletal remains is already good, it clearly states that the remains can be attributed to modern Homo sapiens, I have just added more clarification to the use of anatomical terms. I have – based on the requests from several reviewers and your arguments following this - made a couple of changes where you use the term “skeleton” with e.g., remains.

This is because I agree with the argument that the specimen is indeed only a partial skeleton but also I can see why you would argue for the “most complete” claim as there are indeed not that many postcranial elements for Asian fossils. Where I made the suggestions, I am just toning down the term most complete/skeleton where it is not related to the argument that this is the most complete set of remains in East Asia/Southern China. I am glad that the authors also have highlighted the combination of cranial characteristics with associated postcranial remains.

I will also provide a cleaned-up version of the manuscript with all the changes accepted to make the new suggested shape of it more visible and keeping track all the changes suggested.

AR: We appreciate the suggestions for revision, and we made the revisions accordingly. We agree that the “most complete” is not the best term to describe the skeletal remains; therefore, we made a title change and toned down the “most complete/skeleton”. We state that Liujiang is the best-preserved late Pleistocene human fossil in East Asia/Southern China, in agreement with the recommendations here.

Reviewer #6 (Remarks to the Author):

Modern human skeleton fossil from the Tongtianyan Cave, Liujiang is one of the most complete *H. sapiens* skeleton fossil findings in southern China. The chronology of human fossil from this site is important for understanding the dispersal history of modern humans in East Asia. Previous chronological works on this site suggested an early arrival of modern human in East Asia, but these ages were controversial. This study establishes new chronologic framework for this key site using OSL, U-series and radiocarbon dating, and suggest a younger age for human fossil at Liujiang in correspondence with ages of other human fossils in northern China. The human fossil at Liujiang itself was dated and its provenance in the stratigraphy of the cave is assessed using a comprehensive approach. The paper is well-written and scientifically sound; experiments and results are robust and are clearly present. I think the findings of this paper will attract broad interest for readers of Nature Communication. An earlier version of the manuscript has been submitted to Nature and the authors have properly addressed the comments of four reviewers. Therefore, I think this manuscript is already of high quality, and recommend acceptance of this paper after minor revision. Some minor comments are listed below.

AR: We appreciate the series of positive comments by Reviewer #6 and we are pleased that they value the scientific approach taken in our manuscript. Again, this reviewer points out the international significance of Liujiang human fossils, in conflict with the views of Reviewer #2.

Main text:

Reviewer's comment 1 (RC): Lines 70-71: The chronologies of modern human remains from the Tianyuan Cave and the Zhoukoudian Upper Cave listed here are slightly different from those listed in the abstract. According to Shang et al., 2007, PNAS and Li et al., 2018, JHE, bone samples from human fossil-bearing layers of the Tianyuan Cave and the Zhoukoudian Upper Cave were dated to 42-39 ka cal BP and 38.3-35.8 ka cal BP, respectively. Please check these literatures and use consistent numbers in the manuscript.

AR: Done. The ^{14}C ages of the human remains from the Tianyuan Cave and the Zhoukoudian upper Cave were updated in our revised text. These ^{14}C ages have also been re-calibrated based on the INTALCAL 20 curves as requested by the Nature referees. This is why the age cited was different from the published dates.

Line 80: I suggest the authors cite some references here, regarding the provenance and dating of the Liujiang hominin fossils.

AR: Done. References have now been cited (Main text Line 80).

Lines 81-198: These paragraphs are very informative and I can get information regarding the investigation history of the Tongtianyan Cave, previous dating works of the cave, features of human fossils, geomorphologic setting and stratigraphy of the cave (and potential location of human fossils) by reading through these paragraphs. But to make them easier to follow, I suggest the authors may consider re-order these paragraphs. Personally, I would use such a structure: 1) the location and geomorphologic setting of the Tongtianyan Cave; 2) a brief

investigation history of the cave; 3) the stratigraphy of the cave, archaeological findings of different stratigraphic units and potential location of human fossils; 4) features of human fossils; and 5) previous dating works and their shortage. I reiterate that these paragraphs are already adequately informative, so this structure issue is just a personal advice.

AR: Thank you very much for your suggestions. In our revised manuscript, these paragraphs have been reorganized. To take full account of the comments and suggestions of the referee, some section titles were added, improving the flow and content of our manuscript (Main text Lines 60, 155-174, 175, 206, 243, 338).

You may consider swapping Figure 2 and Figure 3, as in this manuscript Figure 3 (first referred to in Line 163) is mentioned earlier than Figure 2 (first referred to in Line 213).

AR: Done. The order and title of the two figures have been changed (Main text Lines 199-204 & 236-241).

Figure 3: There is no legend for Fig. 3b and Fig. 3c, and I can't see the stratigraphic drawings in Figs. 3b and 3c very clearly. The authors may consider plot Fig. 3b and Fig. 3c separately and add legends in the figures.

AR: Done. Legends for the Fig.3b and Fig.3c were revised (Fig.2b & 2c, Main text Line 199).

Lines 345-347: I can't understand this sentence. Please rephrase.

AR: Done. It has been rephrased (Main text Lines 357-359).

Lines 355-357: Are there any other well-dated sites on this dispersal route supporting this statement?

AR: Here we propose a possible rapid dispersal across Eurasia around 30 ka based on the similarity of cranial shape between Liujiang and Cro-Magnon. Owing to the scarcity of well-preserved human skulls with similar chronologies, we still cannot determine how the dispersal happened, as it may have taken place along the "southern route" via India, or the "northern route" across the Central Asia. As more and more human fossils would be discovered in the future, this dispersal may be better documented. We provide a more cautious statement: "The revised age estimate for Liujiang corresponds with well-known fossil at Cro-Magnon (France, 31-33 ka)³⁴. In addition to chronological contemporaneity, the close affinity of cranial shape between Liujiang and Cro-Magnon 1 from Western Europe^{17,28} imply little morphological differentiation among modern human populations in Europe and East Asia, or rapid dispersal events of early modern humans across Eurasia continent around ~30 ka" (Main text Lines 368-373).

Methods

Line 659: Rb is another ionising irradiation source, although typically with a low content in sediments.

AR: This is a good point. The sentence was revised: "The luminescence emitted from minerals (e.g., quartz and feldspar) under artificial light exposure is proportional to the absorbed energy accumulated within the crystal lattice of minerals by ionizing radiation (e.g., alpha, beta or gamma radiation) from radioactive elements such as uranium (U), thorium (Th), Rubidium (Rb)

and potassium (K) in the environment, as well as cosmic rays” (Main text Lines 714-719).

Line 661: this sentence may be changed to “the total radiation dose absorbed by mineral grains over the burial time...”

AR: Thank you for the suggestion. This sentence been changed accordingly (Main text Lines 719-721).

Line 684: which sample/samples was/were etched for a shorter time?

AR: Done. The samples with less-time etching have been clarified (Main text Lines 743-746).

Line 692: Reference 85 is not about OSL reader. Please check the references and make sure they are cited properly.

AR: Done. It has been corrected (Main text Line 753).

Line 697: have you calibrated your reader using new batches of Riso calibration quartz (Batch 126 and onwards)? If yes, how large is the difference between the apparent machine dose rate obtained from the new batch and the 108 batch? Does it validate the usage of an 8.25% correction factor?

AR: We have re-calibrated our reader using the new batch of RISO calibration quartz (batch 126 of 6.00 Gy) as required by the Nature reviewer and found an underestimation around 8.2%. Considering the possible measurement error, we corrected our previous results using a correction factor of 8.25% as suggested by Autzen et al. (2022).

Lines 701-702: This sentence is unclear. Do you mean the repeated regenerative dose for recycling ratio measurement?

AR: Done. The sentence has been rephrased: “After the standard SAR protocol, a repeated regenerative dose measurement was applied to detect the recycling ratio measurement”. (Main text Lines 761-763)

Lines 704-705: revise to a phrase like “...after preheat and prior to blue light stimulation” and remove repetitive words “with IR stimulation carried out prior to blue stimulation”.

AR: Done. The sentence has been revised: “In addition, to test the purity of the quartz extracts, an additional recycling step was given at last to each aliquot, in which IR stimulation of grains for 40 s at room temperature using infrared LEDs was set after preheat and prior to blue light stimulation” (Lines 763-766)

Lines 708-710: This sentence is not very clear, please rephrase. How long time did you use for OSL measurement?

AR: Done. This has been rephrased: “These measured aliquots having the recycling ratio or OSL IR depletion ratio exceeding the acceptable range (0.9-1.1), or with the recuperation over 5%, were excluded from the D_e determination”. (Main text Lines 768-771)

Line 713: At the end of this paragraph please clarify how did you estimate the mean D_e values for your samples.

AR: Done. The method for mean D_e estimation is clarified in our revised text as follows: "...and the arithmetic mean was used for D_e calculation of each sample". (Main text Lines 773-774)

Lines 719-720: Do you mean 40°C lower than preheat temperatures?

AR: Yes. The cut-heat temperature was set to be 40 °C lower than preheat temperatures. In our revision, the sentence has been changed: "...with the cut-heat ranged from 160 °C to 260 °C at 20 °C increments basically tracking and lagging the preheat temperatures by a margin of 40 °C". (Main text Lines 778-782)

Line 732: Finally in this paragraph, please clarify what are the final preheat and cutheat temperatures used for dating.

AR: Done. A sentence was added: "Finally, a preheat temperature of 260 °C and a cut-heat at 220 °C were used for D_e determination of all samples." (Main text Lines 793-794)

Line 749: I guess you mean "reported by Olley et al"? And I am confused here.... In the supplementary materials you say that the program by Murray et al., (2021) was used for dose rate calculation; in their program, the conversion factors are derived from Guerin et al. (2011) rather than Olley et al. (1996). So, which conversion factors did you use exactly for dose rate estimate? Or do you mean you applied the conversion factors of Olley et al. (1996) for field gamma spectrometry data analysis? Please clarify.

AR: Yes. It has been re-clarified: "Each sample was measured for 8 hours in the field. Spectral data were converted to activity concentrations and infinite matrix dose rates using conversion data by Guerin et al.⁸⁵" (Main text Line 809-811)

Line 760: Reference 95 is absent in the reference list of the main text, and that reference in the Supplementary materials is not related to OSL dating. Please check the cited reference.

AR: Done. The reference has been replaced with "Bøtter-Jensen, L., et al. Developments in radiation, stimulation and observation facilities in luminescence measurements. *Radiation Measurements* 51, 1023-1045 (2003)".

Supplementary Materials

Lines 49-67: There are some repetitions between this paragraph and the main text. Please remove repetitive sentences.

AR: Done. The repetitive sentences have been removed from the Supplementary Information.

Line 184: "Extended data Fig. 2"—I guess you mean Supplementary Fig. 2

AR: Done. It has been changed as "Supplementary Fig.2". (Supplementary Information Line 167)

Supplementary Fig. 2: In figures or in the caption, please indicate which layers are these samples from. For each panel, please use different colors or line styles to differentiate samples.

AR: Done. In the caption of Supplementary Fig.2, the information about the location of these samples are addressed as follows: "Samples of HED-640, HED-641, HED-628 and 21LJ_16 are from Layer 2, HED-627, HED-642, 21LJ_12, 21LJ_14 and 21LJ_15 are from Layer 3,

HED-682 is from Layer 4, HED-683 and HED-684 are from Layer 5 in Unit III, while samples of HED-626, HED-638, HED-639, LJ_13, LJ_17 and LJ_20 are collected from Unit II.” (Supplementary Information Lines 198-202)

Line 438: It should be “In Steps 2 and 6”

AR: Done. It has been changed to “In Steps 2 and 6”. (Supplementary Information Line 426)

Line 439: response to a regenerative dose

AR: Done. It has been revised. (Supplementary Information Line 427)

Line 441: remove “by”

AR: Done. It has been removed in our revised text. (Supplementary Information Line 429)

Supplementary Table 5, step 1: the superscript of D_i should be “a,b”

AR: Done. It has been revised to “a, b” (Supplementary Table 5).

Line 443: I think here you mean “to monitor the efficacy of sensitivity change correction” or “to measure the recycling ratio”

AR: Done. The sentence has been revised into “Repeated dose was given at last to monitor the efficiency of sensitivity change correction”.

Line 461: “the preheat plateau test results...”

AR: Done. It has been revised. (Supplementary Information Line 452)

Line 489: I guess you excluded aliquots with recuperation values $>5\%$

AR: Done. It has been revised to “ $>5\%$ ”. (Supplementary Information Line 480)

Line 491-502: The authors take $2D_0$ values as the upper dating limit for their samples. This upper dating limit estimate, however, is debatable as it is empirical and is based on single saturating exponential growth curve fitting. There is no mathematic validation that D_e values greater than $2D_0$ must be inaccurate—although D_e estimates at higher doses can be less precise due to the flatter growth curve at higher dose range, reliable D_e can be obtained if the shape of the growth curve can be accurately established and if untruncated natural signals are properly used for D_e estimate (e.g. Li et al., 2017). If the growth curves of the samples have an additional linear or exponential component, there is a potential to obtain accurate D_e estimates at higher doses (e.g. Murray et al., 2008; Fu et al., 2017), and for these growth curves the $2D_0$ estimate is inaccurate. The authors use $2D_0$ values to represent the minimum ages for three samples from Unit II. I suggest you use the real measured D_e values to estimate the minimum ages for these samples. Even though these D_e values can be imprecise, these real measured D_e values may not necessarily be wrong (but you can prudently treat them as minimum D_e values), and they can at least provide a more useful lower bound on the true D_e value than the $2D_0$ value (see discussions of Galbraith and Roberts, 2012).

AR: Thank you for the suggestions. In our revised manuscript, we recalculated the D_e values for these three samples from the Unit II by fitting their regeneration data with the sum of an

exponential and a linear component and subsequently calculated the corresponding ages. In our revised manuscript, some clarifications about our new fitting methods were made as follows: “Some studies suggested that reliable D_e can be obtained when the shape of the growth curve can be accurately established and the untruncated natural signals are properly used for D_e estimate¹¹⁰, and it has also been proved that accurate D_e estimates at higher doses¹¹¹⁻¹¹⁴ can actually be obtained by a by a function which is the sum of a single saturating exponential and linear component or by the sum of two saturating exponentials when the growth curves of the samples have an additional linear or exponential components. Thereafter, the data for these three samples, thereafter, were fitted with the sum of an exponential and a linear component, which seem to aligns with the regeneration data points better.” (Supplementary Information Lines 488-499)

Some discussion about the results of these samples from Unit II are also made in our revised text as follows: “However, the three samples from the Unit-II beneath the thick flowstone layer (FL-2) generate the ages of 62 ± 4 ka, 103 ± 7 ka and 151 ± 12 ka, respectively, which are much older than the samples from Unit III. Actually, the samples HED-626 and HED-638 are collected between the flowstone Layer FL-2 and Layer FL-3, which were dated to 71 ± 5 ka and 133 ± 11 ka by carbonate U-series dating method, respectively. While the sample HED-639 are taken from the layer under FL-3 but above the flowstone Layer FL-4 which have an age of 240 ± 37 ka. It can be seen that these ages agree with our as well as previously published U-Th dating results on these flowstone layers 3, even considering the possible age underestimation due to the U-series disequilibrium U-series disequilibrium in this unit.” (Supplementary Information Lines 594-604)

Line 491: 2D0 represents 86% of the saturation intensity

AR: Done. It has been clarified in our revised text: “2D0 (86% of the saturation intensity for a single saturating exponential growth curve fitting) values is generally taken as the practical upper limit for dose estimation”. (Supplementary Information Line 482)

Line 491: “monomolecular fit”—I guess you mean “single saturating exponential growth curve fitting”

AR: Done. It has been revised. (Supplementary Information Line 482)

Lines 492-493: this sentence is not very clear, please rephrase

AR: Done. We rephrased the sentence: “2D0 (86% of the saturation intensity for a single saturating exponential growth curve fitting) values is generally taken as the practical upper limit for dose estimation⁸⁰ to evaluated whether the OSL signals for each measured aliquot is saturated or not, and the aliquots with D_e values higher than which was mostly used to calculate minimum ages.” (Supplementary Information Lines 482-486)

Lines 494-495: It appears to me that your aliquots have very different D0 values. As you used fine grains for dating, I expect the growth curves of different aliquots should be very similar due to strong average effect. Why the D0 values are so different?

AR: The growth curves of different aliquots for these three samples from Unit-II are rather similar, as shown by the standard growth curves after least square-normalisation yielded by the

regeneration data of samples HED-626, HED-638 and HED-639. In our revised text, we removed the sentence and rephrased the sentences as follows: “ $2D_0$ (86% of the saturation intensity for a single saturating exponential growth curve fitting) values is generally taken as the practical upper limit for dose estimation⁸⁰ to evaluated whether the OSL signals for each measured aliquot is saturated or not, and the aliquots with D_e values higher than which was mostly used to calculate minimum ages.” (Supplementary Information Lines 482-486)

R-Fig.1 The standard growth curves (SGC) after least square-normalisation yielded by these regeneration data for the samples HED-626, HED-638 and HED-639.

Line 499: “infinite ages”—I guess this should be “minimum ages”

AR: Done. It has been changed to “minimum ages”. (Supplementary Information Line 485-486)

Lines 503-520: Typically, the OD value and D_e distribution pattern are used for assessing the completeness of OSL signal resetting for single-grain OSL dating. But since this study uses fine grain (4-11 μm) quartz for OSL dating, the apparent OD value for each sample may not represent genuine between-grains difference in D_e value, as in each aliquot there can be tens of thousands of grains contributing to the OSL signal so any between-grain difference in D_e values is averaged. This explains why you got small OD values and very concentrated D_e distribution for all your samples. I think for fine grain dating the OD value and D_e distribution pattern provide little useful information about signal resetting, but the $D_e(t)$ plots, as provided by the authors in their rebuttal letter to the reviewers of Nature, can provide evidence that the OSL signals are sufficiently bleached, and the consistency between OSL ages and radiocarbon ages also suggest good signal bleaching. The authors may consider adding the $D_e(t)$ plots (R-Fig.2) into the Supplementary Information.

AR: Thank you for these constructive suggestions. It is correct that the OD value and D_e distribution pattern make more sense for evaluating the completeness of OSL signal resetting for single-grain OSL dating rather than fine-grain quartz OSL dating. However, in this study, it is used to provide one part of our evidence for good signal resetting of the fine-grain quartz. Besides, the sedimentary processes, and the $D_e(t)$ plots also provide some other supports for our conclusion. Based on your comments, we included the $D_e(t)$ plots in the Supplementary

Information (Supplementary Fig. 13) and made further discussion about the signal resetting issues. We rephrased the paragraph as the follows: “Incomplete resetting is an issue for OSL dating. However, signal bleaching seems not to be a problem for the samples from Tongtianyan cave, for the following reasons: (1) The sediments in Layers 2 and 3 are mainly dominated by homogeneous silty clay, suggesting that they may have been extensively weathered and transported into the cave by low-energy slope sheet flow rather than by flood-like water current. Given that the quartz mineral can be bleached in a few minutes, and the luminescence signals of quartz grains are dominated by the fast component, it therefore seems that these quartz grains may have experienced full exposure to sunlight. Afterwards, owing to the sloping ground surface toward the deep cave in the passage connecting the cave entrance with the chamber where the sediments accumulated, with a dip angle of $\sim 10^\circ$, the sediments entering the cave can be transported smoothly to the deep chamber and trapped there without disturbance. As the deep cave where the sediments deposited is absolutely in darkness, there is no chance for resetting of the OSL signal for them, the burial age of the human fossils seems to be close to that of the resetting time of the luminescence signals outside the cave. (2) All (unsaturated) samples from the Unit-II in Tongtianyan cave typically describe a normal Gaussian distribution of D_e values, with only a few samples slightly skewed (Supplementary Fig. 12), and all give over-dispersion (σ_{OD}) values representing the relative standard deviation of the D_e distribution of $<20\%$, indicating that these sample may have been well bleached prior to deposition^{115,116}. (3) D_e as a function of illumination time ($D_e(t)$), is also used to identify the partial resetting of quartz OSL signals¹¹⁷. As shown in the $D_e(t)$ plots (Supplementary Fig.13), all of the eight unsaturated samples yielded consistent $D_e(t)$ values at the 1σ confidence level independent of signal integration, which also well suggests full resetting of the fast and medium components prior to deposition¹¹⁷. In addition, the consistency between OSL ages and radiocarbon ages in the same layer also provides additional support for the conclusion that there is good OSL signal bleaching for the quartz from Tongtianyan cave.” (Supplementary Information Lines 500-526)

Line 524-548: For samples exhibiting U-series disequilibrium, it would be better to estimate the impact of U-series disequilibrium on dose rate quantitatively, if possible.

AR: Done. In our revised manuscript the possible impact of U-series disequilibrium on dose rate and ages for the three sample from the Unit II were evaluated and are clarified as follows: “These additional ^{238}U resulted from the downward leaching carbonate may have led to an overestimation of the total dose rate for the three sample HED-626, HED-638 and HED-6329 from the Unit II by 0.27 ± 0.27 Gy/ka, 0.94 ± 0.36 Gy/ka and 0.55 ± 0.26 Gy/ka in maximum, respectively, and thus caused an age underestimation of $4\pm 4\%$, $18\pm 8\%$ and $15\pm 8\%$, respectively.” (Supplementary Information Lines 556-560)

Supplementary Fig. 13: It would be great if you could indicate which sample is from with unit/layer in this figure.

AR: Thanks for the suggestion. We added the Unit/Layer name for each sample in our revised figure (Supplementary Fig. 14)

Supplementary Table 7 and Table 8: The errors of OSL ages present in Supplementary Table 7 (given as 1σ error) and Supplementary Table 8 (given as 2σ error) are the same, so I guess these

errors are random-only errors (if the authors use random-only errors in the Bayesian age model). It would be great if the authors could provide random plus systematic error for each age in Supplementary Table 7, which would represent full uncertainty for each age estimate.

AR: The age errors presented in Supplementary Table 7 represent full uncertainties in their age estimations. Except for the random-only errors, these age uncertainties for them also include not only the systematic error yielded during the measurements of radionuclides concentrations and associated dose rate calculations, but also a 2.1% systematic error to allow covering any bias associated with calibration of the laboratory beta source.

In our Bayesian modelling using the Oxcal online program, not only the OSL ages but also the radiocarbon and U-series dates were used. Since the radiocarbon and U-series dating data both contain independent systematic errors and share systematic errors with each other, it is thus necessary to consider not only random errors but also systematic errors in our models. Our results actually show an overall A indices value of 113.7, suggesting that there is no sign of any significant systematic error in the OSL age estimates (Rhodes et al., 2003), which also provide some supports for our conclusions.

Cited References

Autzen et al., "Calibration quartz: An update on dose calculations for luminescence dating", *Radiation Measurements* 157, 106828, 2022

Rhodes, E., Ramsey, C.B., Outram, Z., Batt, C., Willis, L., Dockrill, S., Bond, J., 2003. Bayesian methods applied to the interpretation of multiple OSL dates: high precision sediment ages from Old Scatness Broch excavations, Shetland Isles. *Quaternary Science Reviews* 22, 1231-1244.

Reviewers' Comments:

Reviewer #2:

Remarks to the Author:

My opinion of this paper is unchanged from the last time I reviewed it. Although it is an excellent piece of research that clearly establishes the age of the Liujiang partial skeleton as between ~33 and 24 ka, I still maintain that it is not appropriate for Nature. This is because Nature's reputation stands in publishing material that is game-changing, ground-breaking and clearly of international significance. Unfortunately, the reporting of another *H. sapiens* skeleton from East Asia that is ~33-24 ka in age is no longer in that category – there are already older examples such as Tanyuandong and Upper Cave in China, and Salkhit in Mongolia, as well as various other earlier examples from SE Asia such as Niah and Lida Ajer. Additionally, if the Liujiang remains are as young as 24 ka, they are even less significant internationally.

Liujiang was only important because earlier – and now shown to be erroneous – estimates of its age as early Upper or even late Middle Pleistocene helped fuel arguments for a multi-regional model of human evolution. Even before the re-dating of Liujiang, that model had been almost wholly abandoned in favour of an African origin of *H. sapiens*. With the new dating of Liujiang, it is now in the category of an interesting but not very significant discovery at an international level.

As I suggested previously, the re-dating and skeletal descriptions are certainly worth publication in a journal such as *Journal of Human Evolution*, where the new evidence can be read by the relatively small community familiar with the human skeletal record of China and SE Asia.

Reviewer #5:

Remarks to the Author:

Dear Authors,

Thank you very much for your detailed response to my suggestions and clear layout of the steps you took to integrate the suggestions into your manuscript.

I think the manuscript does flow better now and the message/story of the work is clear and I am happy to recommend it ready for publication.

I noticed these spelling errors, I am sure they can be fixed quickly:

Page 6 line 150 mongloid, should be mongoloid?

Page 17 line 384 pyriform aperture should be piriform aperture

Reviewer #6:

Remarks to the Author:

The authors have addressed all my comments properly. I am happy to support publishing this paper in *Nature Communication*.

Just a few minor points to be corrected:

-Line 744 in the main text: "except for some a sample" should be revised to "except for some samples".

- Line 763 in the main text: "to detect the recycling ratio measurement" may be revised to "to determine the recycling ratio"
- reference #111 in the supplementary materials is not about luminescence dating. Please cite the right reference.

Responses to Reviewers' Comments

We appreciate the editor's and reviewers' constructive comments and suggestions on the previous version of our manuscript, which have been addressed point-by-point as follows.

Reviewer #2 (Remarks to the Author):

My opinion of this paper is unchanged from the last time I reviewed it. Although it is an excellent piece of research that clearly establishes the age of the Liujiang partial skeleton as between ~33 and 24 ka, I still maintain that it is not appropriate for Nature. This is because Nature's reputation stands in publishing material that is game-changing, ground-breaking and clearly of international significance. Unfortunately, the reporting of another *H. sapiens* skeleton from East Asia that is ~33-24 ka in age is no longer in that category – there are already older examples such as Tanyuandong and Upper Cave in China, and Salkhit in Mongolia, as well as various other earlier examples from SE Asia such as Niah and Lida Ajer. Additionally, if the Liujiang remains are as young as 24 ka, they are even less significant internationally.

Liujiang was only important because earlier – and now shown to be erroneous – estimates of its age as early Upper or even late Middle Pleistocene helped fuel arguments for a multi-regional model of human evolution. Even before the re-dating of Liujiang, that model had been almost wholly abandoned in favour of an African origin of *H. sapiens*. With the new dating of Liujiang, it is now in the category of an interesting but not very significant discovery at an international level.

As I suggested previously, the re-dating and skeletal descriptions are certainly worth publication in a journal such as *Journal of Human Evolution*, where the new evidence can be read by the relatively small community familiar with the human skeletal record of China and SE Asia.

AR (authors' reply): Thank you so much for your positive comment on our main contribution: “clearly establishes the age of the Liujiang partial skeleton as between ~33 and 24 ka”. Additionally, we have mentioned in the last letter that the Liujiang human fossils have gained international significance as a representative of early modern humans in East Asia. With these new dates we obtained in this study, the Liujiang human fossils now serve as important material for understanding the dispersal of modern humans across Eurasia in the Later Pleistocene.

Reviewer #5 (Remarks to the Author):

Dear Authors,

Thank you very much for your detailed response to my suggestions and clear layout of the steps you took to integrate the suggestions into your manuscript. I think the manuscript does flow better now and the message/story of the work is clear and I am happy to recommend it ready for publication.

AR: Thank you so much for your positive comments on our work!

I noticed these spelling errors, I am sure they can be fixed quickly:

Page 6 line 150 mongloid, should be mongoloid?

AR: You are right, it has been corrected in our revised manuscript.

Page 17 line 384 pyriform aperture should be piriform aperture

AR: Thank you, it has been corrected in our revision.

Reviewer #6 (Remarks to the Author):

The authors have addressed all my comments properly. I am happy to support publishing this paper in Nature Communication.

AR: Thank you so much for your positive comments on our work!

Just a few minor points to be corrected:

-Line 744 in the main text: “except for some a sample” should be revised to “except for some samples”.

AR: Thank you, it has been corrected into “except for samples”.

-Line 763 in the main text: “to detect the recycling ratio measurement” may be revised to “to determine the recycling ratio”

AR: OK, corrected.

- reference #111 in the supplementary materials is not about luminescence dating. Please cite the right reference.

AR: Thank you, this reference has been replaced by “Fu, X., Cohen, T. J. & Arnold, L. J. Extending the record of lacustrine phases beyond the last interglacial for Lake Eyre in central Australia using luminescence dating. Quaternary Science Reviews 162, 88-110 (2017).” in our revised Supplementary Information.